# GTPO AND GRPO-S: TOKEN AND SEQUENCE-LEVEL REWARD SHAPING WITH POLICY ENTROPY

## ABSTRACT

Reinforcement learning (RL) is a pivotal task for enhancing Large Language Model (LLM) reasoning. Conventional algorithms, however, typically adhere to a coarse-grained credit assignment paradigm, applying a uniform reward to all tokens in a sequence—a critical flaw in long-chain reasoning tasks. In this paper, we address this challenge and propose **Dynamic Entropy Weighting**, a novel mechanism that facilitates fine-grained rewards through two new algorithms: **Group Token Policy Optimization (GTPO)**, which assigns an entropy-weighted reward to each token, and the analogous algorithm **Sequence-Level GRPO (GRPO-S)**. Our approach is founded on the hypothesis that high policy entropy within a reasoning path is a powerful heuristic for "cognitive effort" at pivotal junctures, which can be repurposed into a learning signal. By repurposing policy entropy for reward shaping, we achieve true per-token credit assignment. Experimental results across challenging reasoning benchmarks validate the superiority of our approach, showing our methods significantly outperform a strong DAPO baseline and confirming our entropy-weighting mechanism as the key driver of this performance boost.

## 1 INTRODUCTION

The reasoning capabilities of Large Language Models (LLMs) have evolved profoundly, transitioning from pattern recognition to simulating deeper cognitive processes (Guo et al., 2025; Wei et al., 2022; Zhang et al., 2022; Achiam et al., 2023). This leap is evidenced by state-of-the-art performance on formidable tasks like advanced mathematics and competitive coding, where models learn complex, process-oriented behaviors such as self-verification and iterative refinement (Chen et al., 2025; Lewkowycz et al., 2022; Li et al., 2022; Madaan et al., 2023). At the heart of this paradigm shift lies large-scale Reinforcement Learning (RL), the core technology enabling models to acquire dynamic, multi-step problem-solving strategies beyond the scope of static supervised learning (Zhang et al., 2025b; Wang et al., 2016; Schulman et al., 2017; Ouyang et al., 2022).

This evolution in alignment algorithms has followed a clear trajectory: a persistent quest for greater simplicity and efficiency (Mnih et al., 2015; Amini et al., 2024). Early Reinforcement Learning from Human Feedback (RLHF) pipelines, often built on Proximal Policy Optimization (PPO), were powerful but notoriously cumbersome, necessitating separate reward and value models that introduced computational overhead and training instabilities (Schulman et al., 2017; Christiano et al., 2017; OpenAI Spinning Up, 2018; Gao et al., 2023). In response, the field gravitated towards more direct paradigms. Direct Preference Optimization (DPO) marked a milestone by bypassing explicit reward modeling (Rafailov et al., 2023; 2024; Lai et al., 2024). This trend culminated in value-function-free methods like Group Relative Policy Optimization (GRPO), which simplifies the optimization landscape by using a group's average reward as an advantage baseline, making it ideal for fine-tuning massive models ((Shao et al., 2024; Yu et al., 2025); see Appendix B for a detailed review of alignment algorithm evolution).

However, this pursuit of efficiency has created a critical bottleneck: **coarse-grained credit assignment** (Wei et al., 2022). Algorithms like GRPO assign a uniform reward to every token in a sequence based solely on the final outcome (Shao et al., 2024). This limitation becomes profound in tasks requiring long-chain reasoning. For instance, a sequence with dozens of correct logical steps may receive zero reward for a single final error, penalizing correct and incorrect reasoning alike

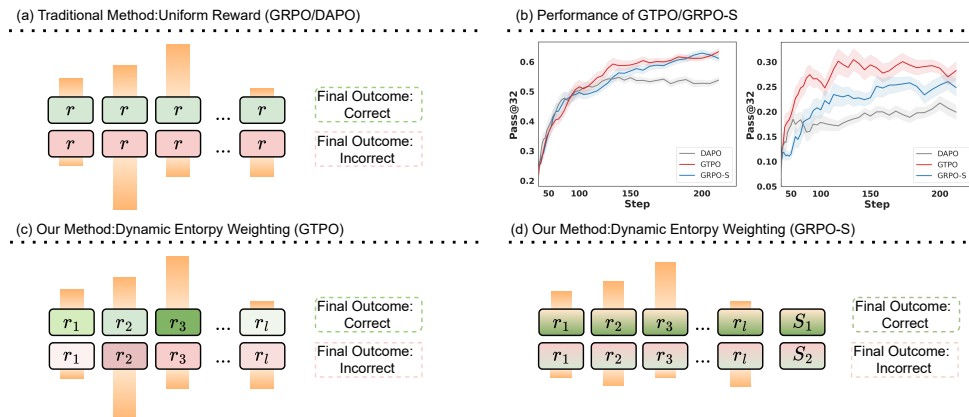

**Figure 1:** Conceptual illustration of reward assignment. (a) Traditional methods assign a uniform reward based on the final outcome. In contrast, our methods use Dynamic Entropy Weighting to refine credit assignment: (c) GTPO rewards high-entropy tokens in correct sequences while suppressing them in incorrect ones, and (d) GRPO-S rewards correct sequences with higher average entropy while penalizing incorrect paths. Yielding superior performance (b).

(Yang et al., 2025; Wang et al., 2025). Conversely, a sequence that reaches a correct answer through flawed or guessed intermediate steps is fully rewarded (Sutton and Barto, 2018; Zhang et al., 2025a). This sparse, imprecise feedback constrains learning, creating an efficiency-precision trade-off where imprecise credit assignment is the primary barrier to progress (Bansal et al., 2023).

To distinguish the quality of steps in answers and address the coarse-grained credit assignment bottleneck, this paper uses policy entropy to increase the rewards for key reasoning steps in correct answers and reduce the suppression of effective exploration signals in incorrect answers. The central hypothesis is that moments of high policy entropy within a reasoning sequence are not random noise but strong correlates of pivotal reasoning junctures (Cheng et al., 2025; Cui et al., 2025). When a model selects between multiple valid mathematical theorems or constructs a complex logical connective, its uncertainty—as measured by entropy—naturally increases. This policy entropy, traditionally viewed as a measure of model indecision, can be repurposed as a powerful heuristic for cognitive effort (Haarnoja et al., 2018; Lindsay, 2020). In successful paths, it signals a moment of valuable exploration to be reinforced; in unsuccessful paths, it can help the policy break from incorrect thinking. This principle motivates the proposal of **Dynamic Entropy Weighting**, a novel framework that reshapes the reward signal to be proportional to token-level or sequence-level entropy, thereby focusing the policy gradient on the most critical decision points.

This principle is operationalized through a suite of two complementary algorithms. The first, **Group Token Policy Optimization (GTPO)**, is a novel token-level algorithm that assigns a unique, entropy-weighted reward to every token, achieving the first true, fine-grained, per-token credit assignment within the efficient GRPO framework. Complementing this, **Sequence-Level Group Relative Policy Optimization (GRPO-S)** is a lightweight variant that modulates the global reward for an entire sequence based on its average entropy. Together, these methods offer a principled trade-off between granular precision and computational cost, as illustrated in Fig. 1, which conceptually demonstrates how entropy modulation at both token and sequence levels leads to more nuanced credit assignment and improved performance.

This paper makes the following principal contributions:

- We formalize coarse-grained credit assignment as a fundamental bottleneck in value-function-free RL, demonstrating its impact on learning efficiency for long-chain reasoning.

- We propose **Group Token Policy Optimization (GTPO)**, a novel token-level algorithm that introduces a dynamic, entropy-weighted reward mechanism to achieve precise, per-token credit assignment within the efficient GRPO framework.

- We develop **Sequence-Level Group Relative Policy Optimization (GRPO-S)** as an analogous sequence-level algorithm that uses the dynamic, entropy-weighted reward mechanism to capture the exploratory value of an entire sequence.

- We provide a theoretical analysis, rooted in variance reduction arguments, to motivate our objective function design and conduct comprehensive experiments on challenging reasoning benchmarks, showing that our methods significantly outperform strong baselines.

# 2 DYNAMIC ENTROPY WEIGHTING FOR POLICY OPTIMIZATION

This section introduces Dynamic Entropy Weighting, a framework that addresses coarse-grained credit assignment by repurposing policy entropy for fine-grained reward shaping, motivated by the statistical limitations of GRPO (§ 2.1). Two algorithms are presented: the token-level Group Token Policy Optimization (GTPO) (§ 2.2) and the sequence-level GRPO-S (§ 2.3), and conclude with an analysis of implementation and convergence (§ 2.4).

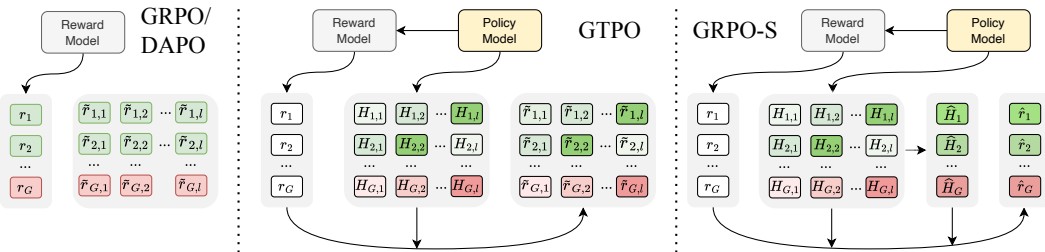

**Figure 2:** A high-level comparison of the reward signaling process. Conventional methods like GRPO/DAPO use a static reward model to assign a uniform reward to an entire sequence. Our framework, encompassing GTPO and GRPO-S, introduces a Dynamic Entropy Weighting module that reshapes this signal into fine-grained rewards at either the token or sequence level before it is used by the policy model.

## 2.1 FROM COARSE-GRAINED CREDIT ASSIGNMENT TO DYNAMIC ENTROPY WEIGHTING

**Background: Group Relative Policy Optimization and Its Limitations.** Our work builds upon the Group Relative Policy Optimization (GRPO) framework (Shao et al., 2024), a value-function-free algorithm that simplifies policy optimization for LLMs. Given a prompt $q$, GRPO samples a group of $G$ sequences, $\{o_1, o_2, \ldots, o_G\}$, from a policy $\pi_\theta$. Each sequence $o_i$ receives a terminal reward $r_i$ (e.g., 1 for correct, 0 for incorrect). The advantage function for all tokens within a sequence $o_i$ is defined as the sequence's reward normalized relative to the group's average reward:

$$\hat{A}_i = \frac{r_i - \text{mean}(\{r_k\}_{k=1}^G)}{\text{std}(\{r_k\}_{k=1}^G)}. \tag{1}$$

The GRPO objective then applies this uniform advantage estimate to every token in the sequence within a PPO-style clipped loss function (Shao et al., 2024). While simple and effective, this uniform application of the advantage is the central mechanism of GRPO, and it also represents the core limitation that motivates our work: **coarse-grained credit assignment**. This approach is not only conceptually imprecise but, as will now be demonstrated, also statistically suboptimal.

**Motivation: The Statistical Case for Finer-Grained Advantage Estimation.** A key motivation for our shift towards a token-level objective stems from a variance reduction argument concerning the baseline term in the advantage function (i.e., the group's average reward). When sequence lengths $|o_i|$ are unequal, there are two primary ways to estimate this baseline:

$$\textbf{Sequence-level mean:} \hat{R}_1 = \frac{1}{G} \sum_{i=1}^G r_i. \qquad \textbf{Token-level mean:} \hat{R}_2 = \frac{\sum_{i=1}^G |o_i| r_i}{\sum_{i=1}^G |o_i|}.$$

While a token-level reward baseline provably reduces variance for more stable gradients ($\text{Var}(\hat{R}_2) \leq \text{Var}(\hat{R}_1)$, see Appendix C.1 for a formal proof), this statistical license is insufficient. To fully exploit this granularity, a principled reshaping of the reward signal itself is essential. This is achieved through dynamic entropy weighting, which, as illustrated in Fig. 2, focuses the policy gradient on critical decision points to create a far more instructive and fine-grained learning signal than conventional methods.

**Solution: The Dynamic Entropy Weighting Framework.** Our framework is built upon the hypothesis that high-entropy moments within a reasoning sequence are not noise but signatures

of pivotal junctures. Policy entropy, traditionally a measure of uncertainty, is repurposed as a heuristic for "cognitive effort", transforming the sparse, binary reward into a dense, fine-grained learning signal. The framework partitions sequences by their terminal reward into successful ($O^+$) and unsuccessful ($O^-$) sets, enabling a dual strategy for credit assignment. High-entropy tokens in successful sequences ($o_i \in O^+$) receive a reward bonus to reinforce valuable exploration. Conversely, low-entropy tokens in unsuccessful sequences ($o_j \in O^-$) are assigned larger penalties to discourage confident but incorrect reasoning. This precise modulation focuses the policy gradient on the most informative steps of the reasoning process.

## 2.2 GROUP TOKEN POLICY OPTIMIZATION (GTPO)

Group Token Policy Optimization (GTPO) is the most direct and granular implementation of our framework. It introduces a fine-grained, entropy-weighted credit assignment mechanism that operates at the individual token level.

**Token-Level Reward Shaping.** For any token $o_{i,t}$ within a **successful sequence** $o_i \in O^+$, $o_{j,t}$ within an **unsuccessful sequence** $o_j \in O^-$, the following entropy-weighted reward is defined:

$$\tilde{r}_{i,t}^+ = \alpha_1 r_i + \alpha_2 \frac{H_{i,t}}{\sum_{k=1}^n H_{k,t}} \cdot d_t \quad \text{and} \quad \tilde{r}_{j,t}^+ = 0. \tag{2}$$

This reward is composed of the original binary success signal $r_i$ (where $r_i = 1$) and a dynamic entropy bonus, balanced by hyperparameters $\alpha_1, \alpha_2 > 0$. The bonus is proportional to the token's generation entropy, $H_{i,t} = -\sum_{v \in \mathcal{V}} \pi_{\theta_{\text{old}}}(v|q, o_{i,<t}) \log \pi_{\theta_{\text{old}}}(v|q, o_{i,<t})$. Crucially, this entropy is normalized across all $n$ successful sequences at timestep $t$, creating a *relative* signal that rewards valuable exploration—tokens generated with higher uncertainty compared to alternative successful paths (if a sequence $o_k$ has length less than $t$, its $H_{k,t}$ is treated as 0). This relative bonus is then scaled by $d_t$, the count of successful sequences with length $\geq t$, which dynamically adjusts the reward magnitude to account for the diminishing number of active reasoning paths over time.

For any token $o_{j,t}$ within an **unsuccessful sequence** $o_j \in O^-$, the goal is to penalize confident mistakes more heavily. Its reward $\tilde{r}_{j,t}^-$ is thus defined using inverse entropy, which assigns a larger penalty to low-entropy (i.e., high-confidence) tokens:

$$\tilde{r}_{j,t}^- = \alpha_1 \cdot (-1) + \alpha_2 \frac{1/H_{j,t}}{\sum_{k=1}^m (1/H_{k,t})} \cdot h_t \cdot (-1) \quad \text{and} \quad \tilde{r}_{i,t}^- = 0, \tag{3}$$

where $h_t$ is the count of unsuccessful sequences with length $\geq t$. This formulation encourages the model to be uncertain when it is incorrect, promoting exploration away from failure modes. Based on these shaped rewards, separate advantage functions are computed for the positive and negative sets, both normalized over all tokens in the entire batch to ensure a consistent scale:

$$\tilde{A}_{i,t}^+ = \frac{\tilde{r}_{i,t}^+ - \text{mean}(\tilde{R}^+)}{\text{std}(\tilde{R}^+)} \quad \text{and} \quad \tilde{A}_{j,t}^- = \frac{\tilde{r}_{j,t}^- - \text{mean}(\tilde{R}^-)}{\text{std}(\tilde{R}^-)}. \tag{4}$$

Here, $\tilde{R}^+$ and $\tilde{R}^-$ represent the collections of all shaped token rewards across all positive and negative sequences in the batch, respectively.

**The GTPO Objective Function.** The final objective function for GTPO integrates these components into a unified, token-level PPO-style loss. The expectation is taken over all tokens in the batch, weighted by the reciprocal of the total number of tokens $1/\sum_{k=1}^G |o_k|$:

$$\mathcal{J}_{\text{GTPO}}(\theta) = \mathbb{E}\Bigg[ \frac{1}{\sum_{k=1}^G |o_k|} \Bigg( \sum_{i=1}^n \sum_{t=1}^{|o_i|} \min\left(w_{i,t}(\theta)\tilde{A}_{i,t}^+, \text{clip}(w_{i,t}(\theta), 1-\epsilon, 1+\epsilon)\tilde{A}_{i,t}^+\right)$$
$$+ \sum_{j=n+1}^G \sum_{t=1}^{|o_j|} \min\left(w_{j,t}(\theta)\tilde{A}_{j,t}^-, \text{clip}(w_{j,t}(\theta), 1-\epsilon, 1+\epsilon)\tilde{A}_{j,t}^-\right) \Bigg) \Bigg], \tag{5}$$

where $w_{i,t}(\theta) = \frac{\pi_\theta(o_{i,t}|q, o_{i,<t})}{\pi_{\theta_{\text{old}}}(o_{i,t}|q, o_{i,<t})}$ is the standard importance sampling weight.

## 2.3 A SEQUENCE-LEVEL VARIANT OF GTPO (GRPO-S)

While GTPO offers maximal granularity, it incurs computational overhead for per-token entropy and reward calculation. Additionally, since some tasks are result-oriented, the goal is to develop a corresponding sequence-level algorithm by following the approach of GTPO, and perform further refinement. Hence GRPO-S is proposed as an analogous method that applies our entropy-weighting principle at the sequence level. The core idea is to modulate the reward for an entire sequence based on its overall exploratory value, as captured by its average entropy.

**Sequence-Level Reward Shaping.** For any sequence $o_k \in O$, rewards are shaped based on a sequence's average token entropy, $\hat{H}_k = \frac{1}{|o_k|} \sum_{t=1}^{|o_k|} H_{k,t}$. For successful sequences ($o_i \in O^+$), the reward is augmented with an entropy-based bonus to reinforce valuable exploration. Conversely, for unsuccessful sequences ($o_j \in O^-$), an additional penalty proportional is applied to their average inverse entropy, thus penalizing high-confidence mistakes more severely. Formally:

$$\hat{r}_i^+ = \beta_1 r_i + \beta_2 \frac{\hat{H}_i}{\sum_{k=1}^n \hat{H}_k} \cdot n \qquad \text{and} \qquad \hat{r}_j^- = \beta_1 \cdot (-1) + \beta_2 \frac{1/\hat{H}_j}{\sum_{k=1}^m (1/\hat{H}_k)} \cdot m \cdot (-1), \quad (6)$$

where $\beta_1, \beta_2 > 0$ are hyperparameters. This formulation rewards successful sequences that are, on average, more exploratory, while penalizing confidently incorrect sequences.

**The GRPO-S Objective Function.** The advantage functions $\hat{A}_i^+$ and $\hat{A}_j^-$ are computed analogously to Equation 1, but using the sequence-level shaped rewards and normalizing over the $G$ sequences in the group. The final objective function for GRPO-S mirrors the structure of the original GRPO loss, but with our shaped advantages:

$$\mathcal{J}_{\text{GRPO-S}}(\theta) = \mathbb{E}\Bigg[ \frac{1}{G} \bigg( \sum_{i=1}^n \min\big(\hat{w}_i(\theta)\hat{A}_i^+, \text{clip}(\hat{w}_i(\theta), 1-\epsilon, 1+\epsilon)\hat{A}_i^+\big)$$

$$+ \sum_{j=n+1}^G \min\big(\hat{w}_j(\theta)\hat{A}_j^-, \text{clip}(\hat{w}_j(\theta), 1-\epsilon, 1+\epsilon)\hat{A}_j^-\big) \bigg) \Bigg], \tag{7}$$

where the sequence-level importance weight $\hat{w}_i(\theta)$ averages the token-level weights:

$$\hat{w}_i(\theta) = \frac{1}{|o_i|} \sum_{t=1}^{|o_i|} w_{i,t}(\theta) = \frac{1}{|o_i|} \sum_{t=1}^{|o_i|} \frac{\pi_\theta(o_{i,t}|q, o_{i,<t})}{\pi_{\theta_{\text{old}}}(o_{i,t}|q, o_{i,<t})}. \tag{8}$$

## 2.4 IMPLEMENTATION AND THEORETICAL GUARANTEES

### 2.4.1 IMPLEMENTATION DETAILS

This section consolidates the practical details required to implement our framework, including a critical mechanism for ensuring theoretical guarantees and a detailed algorithmic procedure.

**Robust Concept Definitions via Geometric Mean.** The arithmetic mean for aggregating sequence-level importance sampling weights can cause training instability. The underlying ratio-based weights have a skewed distribution, making the arithmetic mean sensitive to outliers. To mitigate this, the robust geometric mean is employed. By averaging in logarithmic space, the geometric mean is better suited for ratios and dampens the influence of extreme values, yielding a stable aggregation. The sequence-level importance weight is thus redefined as follows:

$$\tilde{r}_{i,t} = \alpha_1 r_i + \alpha_2 \frac{H_{i,t}}{(\prod_{k=1}^{|o_i|} H_{k,t})^{1/|o_i|}}, \qquad \tilde{r}_{j,t}^- = \alpha_1(-1) + \alpha_2 \frac{1/H_{j,t}}{(\prod_{k=1}^{|o_j|} 1/H_{k,t})^{1/|o_j|}}(-1),$$

$$\hat{r}_i^+ = \beta_1 r_i + \beta_2 \frac{\hat{H}_i}{(\prod_{k=1}^n \hat{H}_k)^{1/n}}, \qquad \hat{r}_j^- = \beta_1(-1) + \beta_2 \frac{1/\hat{H}_j}{(\prod_{k=1}^{|o_j|} 1/\hat{H}_k)^{1/|o_j|}}(-1),$$

$$\hat{H}_k = (\prod_{t=1}^{|o_k|} H_{k,t})^{1/|o_k|}, \qquad \hat{w}_i(\theta) = (\prod_{t=1}^{|o_i|} w_{i,t}(\theta))^{1/|o_i|}.$$

**Algorithmic Procedure.** To clearly illustrate the implementation flow of GTPO and GRPO-S, the complete training procedure is provided in Algorithm 1. The procedure highlights the shared steps and the key differences between the token-level and sequence-level approaches.

---

**Algorithm 1** GTPO and GRPO-S Training Procedure

---

**Initialize:** Policy parameters $\theta$, reference policy $\pi_{\theta_{\text{ref}}}$, hyperparameters $\alpha, \beta, G, \epsilon$.
**for** each training iteration $k = 1, 2, \ldots$ **do**
    Sample a batch of prompts $\{q\}$.
    $\pi_{\theta_{old}} \leftarrow \pi_\theta$
    For each prompt $q$, sample a group of $G$ sequences $\{o_1, \ldots, o_G\}$ using $\pi_{\theta_{old}}$.
    For each sequence $o_i$, compute its terminal reward $r_i \in \{0, 1\}$.
    Partition the group into successful sequences $O^+$ and unsuccessful $O^-$.
    Compute policy entropy $H_{i,t}$ for each token $o_{i,t}$ using $\pi_{\theta_{old}}$.

    **— GTPO Branch (Token-Level) —**
    For each token $o_{i,t} \in O^+$ and $o_{j,t} \in O^-$, compute $\tilde{r}^+_{i,t}$ and $\tilde{r}^-_{j,t}$ using Eq. (2) and (3).
    Compute token-level advantages $\tilde{A}^+_{i,t}$ and $\tilde{A}^-_{j,t}$ using Eq. (4).
    Compute the GTPO loss $\mathcal{J}_{\text{GTPO}}(\theta)$ using Eq. (5).
    Update $\theta$ using a gradient step on $\mathcal{J}_{\text{GTPO}}(\theta)$.

    **— GRPO-S Branch (Sequence-Level) —**
    For each sequence $o_i \in O^+$ and $o_j \in O^-$, compute average entropy $\hat{H}_i$ and $\hat{H}_j$.
    Compute shaped sequence rewards $\hat{r}^+_i$ and $\hat{r}^-_j$ using Eq. (6).
    Compute sequence-level advantages $\hat{A}^+_i$ and $\hat{A}^-_j$.
    For each sequence, compute the geometric importance weight $\hat{w}_i(\theta)$ using Eq. (8).
    Compute the GRPO-S loss $\mathcal{J}_{\text{GRPO-S}}(\theta)$ using Eq. (7).
    Update $\theta$ using a gradient step on $\mathcal{J}_{\text{GRPO-S}}(\theta)$.
**end for**

---

### 2.4.2 THEORETICAL GUARANTEES

This section provides a theoretical analysis establishing that our proposed reward shaping mechanisms preserve the expected policy gradient direction of the baseline algorithms, a key condition for ensuring convergence. Our approach redistributes rewards to create a more granular, token-aware learning signal. As the entropy term is detached from the gradient computation, our analysis demonstrates that while these modifications alter training dynamics, they maintain the expected gradient direction, thus guiding optimization towards a valid policy optimum.

**Analysis of the Token-Level Objective (GTPO).** Our token-level objective modifies the reward structure of the GRPO baseline. As our reward shaping applies exclusively to successful sequences (where the original reward $r_i = 1$), the analysis centers on this redistribution. By construction, the shaping is conservative; the weighted redistribution of token-level rewards $\tilde{r}^+_{i,t}$ for these positive sequences preserves the total reward:

$$\sum_{i,t} \tilde{r}^+_{i,t} = (\alpha_1 + \alpha_2) \sum_{i=1}^{G} |o_i| r_i \approx \sum_{i=1}^{G} |o_i| r_i.$$

The equations hold because $\alpha_1 + \alpha_2 \approx 1$ has been set. The conservation of total reward leads directly to the conclusion that the expected mean reward remains unchanged:

$$\mathbb{E}\big[\text{mean}(\{\hat{r}^+_{k,t}\}_{1 \leq k \leq G,\ 1 \leq t \leq |o_k|})\big] \approx \mathbb{E}\big[\text{mean}(\{r_{k,t}\}_{1 \leq k \leq G,\ 1 \leq t \leq |o_k|})\big],$$

where $\text{mean}(R) = \frac{\sum_i |o_i| r_i}{\sum_i |o_i|}$. This equivalence, in turn, implies that the expectation of the advantage function under our proposed reward shaping, $\mathbb{E}[\tilde{A}^+_{i,t}]$, is approximately equal to the expectation of the baseline advantage function, $\mathbb{E}[\hat{A}_{i,t}]$ ($\mathbb{E}[\tilde{A}^+_{i,t}] \approx \mathbb{E}[\hat{A}_{i,t}]$), where $\hat{A}_{i,t} = \frac{r_i - \text{mean}(R)}{\text{std}(R)}$. The source of this approximation and a detailed description of $\text{mean}(R)$ are provided in Appendix C.4.

Since the entropy term is detached, the expected policy gradient $\mathbb{E}$ approximates that of the GRPO baseline ($\mathbb{E}[\nabla_\theta \mathcal{J}_{\text{GTPO}}(\theta)] \approx \mathbb{E}[\nabla_\theta \mathcal{J}_{\text{DAPO}}(\theta)]$). This preserves the expected gradient direction, ensuring optimization towards the same local optimum. The modification primarily impacts training dynamics by altering the gradient estimator's variance. This redistribution provides a fine-grained, token-level signal that can yield a lower-variance estimator, potentially stabilizing and accelerating

convergence, as detailed in § C.2. A parallel analysis for our sequence-level objective (GRPO-S), which similarly preserves the expected policy gradient while aiming to reduce estimator variance, is provided in Appendix C.3.

## 3 EXPERIMENTS

### 3.1 EXPERIMENTAL SETUP

**Tasks and Datasets.** Methods are evaluated on the **AIME 2024** and **AIME 2025** benchmarks. These challenging mathematical datasets require long-horizon, chain-of-thought reasoning, making them a rigorous testbed for assessing advanced alignment techniques.

**Evaluation Metrics.** Our primary metric is **Pass@k** ($k \in \{2, 4, 8, 16, 32\}$), which encourages solution diversity over the more conservative Pass@1. **Mean@32** is also reported. Improvements are quantified by **Absolute (APG)** and **Relative (RPG) Performance Gains** over the baseline.

**Models and Baselines.** Experiments are conducted on **Qwen2.5-7B** and **Qwen2.5-32B** models. Comparisons are made against faithful implementations of **GRPO** (Shao et al., 2024) and **DAPO** (Yu et al., 2025), both state-of-the-art alignment baselines, to ensure a strong comparison.

**Implementation Details.** Experiments were run on 64 GPUs with a global batch size of 128, a group size of 16, and a learning rate of $1 \times 10^{-6}$. For generation, a temperature 1.0 and top-p 1.0 are used, with max lengths of 2048 (prompt) and 4096 (response). Key reward shaping hyperparameters are $\alpha_1 = \beta_1 = 1$, $\alpha_2 = \beta_2 = 0.1$, and entropy is clipped at $\epsilon_{low} = 0.2$, $\epsilon_{high} = 0.28$.

| AIME 2025* | Mean@32 | Pass@2 | Pass@4 | Pass@8 | Pass@16 | Pass@32 |
|---|---|---|---|---|---|---|
| *Qwen2.5-7B* | | | | | | |
| GRPO | 0.1500 | 0.1606 | 0.1903 | 0.1990 | 0.2000 | 0.2000 |
| DAPO | 0.1333 | 0.1508 | 0.1628 | 0.1663 | 0.1667 | 0.1667 |
| **GRPO-S** | 0.1833 | 0.2093 | 0.2276 | 0.2329 | 0.2333 | 0.2333 |
| **GTPO** | 0.1667 | 0.2201 | 0.2551 | 0.2654 | 0.2667 | 0.2667 |
| **APG of GRPO-S(Vs DAPO)** | +0.0500 | +0.0585 | +0.0648 | +0.0666 | +0.0666 | +0.0666 |
| **RPG of GRPO-S(Vs DAPO)** | +37.5% | +38.8% | +39.8% | +40.1% | +40.0% | +40.0% |
| **APG of GTPO(Vs DAPO)** | +0.0334 | +0.0693 | +0.0923 | +0.0991 | +0.1000 | +0.1000 |
| **RPG of GTPO(Vs DAPO)** | +25.1% | +46.0% | +56.7% | +59.6% | +60.0% | +60.0% |
| *Qwen2.5-32B* | | | | | | |
| GRPO | 0.1771 | 0.2255 | 0.2595 | 0.2727 | 0.2797 | 0.2885 |
| DAPO | 0.2167 | 0.2346 | 0.2576 | 0.2658 | 0.2667 | 0.2667 |
| **GRPO-S** | 0.2511 | 0.2634 | 0.2975 | 0.3156 | 0.3293 | 0.3433 |
| **GTPO** | 0.2689 | 0.3064 | 0.3349 | 0.3588 | 0.3660 | 0.3667 |
| **APG of GRPO-S(Vs DAPO)** | +0.0344 | +0.0288 | +0.0399 | +0.0498 | +0.0626 | +0.0766 |
| **RPG of GRPO-S(Vs DAPO)** | +15.9% | +12.3% | +15.5% | +18.7% | +23.5% | +28.7% |
| **APG of GTPO(Vs DAPO)** | +0.0522 | +0.0718 | +0.0773 | +0.0930 | +0.0993 | +0.1000 |
| **RPG of GTPO(Vs DAPO)** | +24.1% | +30.6% | +30.0% | +35.0% | +37.2% | +37.5% |
| **AIME 2024** | **Mean@32** | **Pass@2** | **Pass@4** | **Pass@8** | **Pass@16** | **Pass@32** |
| *Qwen2.5-32B* | | | | | | |
| GRPO | 0.2917 | 0.3238 | 0.3718 | 0.4161 | 0.4452 | 0.4630 |
| DAPO | 0.3406 | 0.4047 | 0.4664 | 0.5222 | 0.5643 | 0.5902 |
| **GRPO-S** | 0.3552 | 0.4382 | 0.5112 | 0.5750 | 0.6243 | 0.6719 |
| **GTPO** | 0.3521 | 0.4299 | 0.4941 | 0.5781 | 0.6461 | 0.6891 |
| **APG of GRPO-S(Vs DAPO)** | +0.0146 | +0.0335 | +0.0448 | +0.0528 | +0.0600 | +0.0817 |
| **RPG of GRPO-S(Vs DAPO)** | +4.3% | +8.3% | +9.6% | +10.1% | +10.6% | +13.8% |
| **APG of GTPO(Vs DAPO)** | +0.0115 | +0.0252 | +0.0277 | +0.0559 | +0.0818 | +0.0989 |
| **RPG of GTPO(Vs DAPO)** | +3.4% | +6.2% | +5.9% | +10.7% | +14.5% | +16.8% |

**Table 1:** Performance of GTPO and GRPO-S against the DAPO and GRPO baselines on AIME 2024 and 2025 benchmarks, reporting maximum Pass@k and Mean@32 scores. APG/RPG denote Absolute/Relative Performance Gain over DAPO. Our methods show significant, consistent improvements. *Pass@k on AIME 2025 plateaus for k > 4 due to the limited test set size.

### 3.2 COMPARATIVE PERFORMANCE ANALYSIS

As presented in Table 1, our methods, GTPO and GRPO-S, establish a new state-of-the-art by consistently and substantially outperforming both the GRPO and DAPO baselines across all

configurations. Our approach critically resolves a trade-off observed in prior methods: while a strong baseline like DAPO demonstrates competitive performance, its stability-focused loss can limit exploration, causing it to fall short on Pass@k metrics in certain settings (e.g., AIME 2025). In stark contrast, our entropy-based reward shaping successfully elevates both the overall performance metric (Mean@32) and these exploration-sensitive scores. The efficacy of this mechanism is particularly pronounced on smaller models; for instance, GTPO's relative performance gain (RPG) over DAPO on the AIME 2025 benchmark reaches a remarkable +60.0% on the 7B model, compared to +37.5% on the 32B model. This suggests our entropy-weighting provides a crucial learning signal for smaller models, which are more susceptible to premature convergence, by encouraging exploration and helping them navigate complex reasoning spaces more effectively.

### 3.3 REWARD TRAJECTORIES AND SAMPLE EFFICIENCY

The mean reward trajectories on the test set (Fig. 4) illuminate the benefits of our approach, demonstrating that our methods not only achieve a significantly higher final reward ceiling but do so with strong sample efficiency. The models largely converge within **210 training steps**, a finding substantiated by the training set reward curves, which are deferred to Appendix E.4 for brevity. This rapid convergence indicates that the substantial benefits of enhanced exploration do not come at the cost of slower learning. Across all datasets and model sizes, GTPO consistently reaches the highest reward plateau, followed closely by GRPO-S, with both substantially outperforming DAPO.

Our methods foster sustained exploration and prevent policy collapse, establishing a clear causal chain from our entropy-weighted reward to superior performance. A detailed analysis and the empirical evidence for this mechanism are presented in Appendix E.1.

### 3.4 HYPERPARAMETER SENSITIVITY ANALYSIS

We conducted a sensitivity analysis on the key reward shaping hyperparameters for GTPO and GRPO-S, with results presented in Fig. 3. Across all tested configurations, both of our methods demonstrate robust and significant performance gains over the DAPO baseline on Mean@32 and Pass@32 metrics. Among them, GRPO-S exhibits higher stability across settings, providing a clear view of the performance dynamics. As the weight of the entropy bonus ($\beta_2$) increases from 0.1 to 0.2, we observe a clear decline in overall performance for GRPO-S. This finding confirms that while exploration is beneficial, an excessive entropy bonus can detract from optimizing the primary task objective.

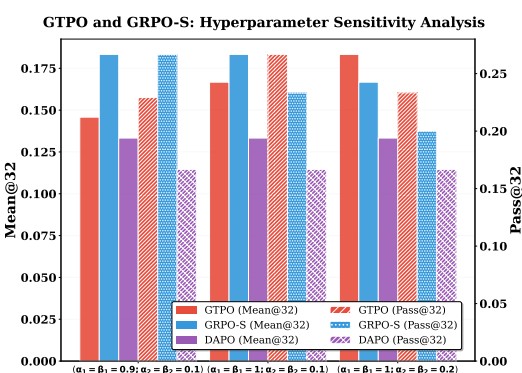

**Figure 3:** Hyperparameter comparison.

## 4 DISCUSSION

**Batch-Level Entropy Comparison as Implicit Curriculum Learning.** Our current implementation operates at the batch level, which compares entropy across different problems, creating an efficient implicit curriculum. The algorithm naturally directs larger gradients towards solvable but high-entropy problems that represent the frontier of the model's capabilities. As the model gains proficiency, the entropy of these problems decreases, causing the learning focus to automatically shift to the next set of challenging tasks. This design leverages the model's own uncertainty as a dynamic signal for learning priority, thus avoiding the need for curriculum design.

**Future Work.** The concept of relative entropy itself warrants deeper exploration. While this work highlights its importance, the optimal method for comparison remains an open question. Three distinct approaches for measuring this relativity are identified. Visualizing the token entropies as a matrix $H$, where $H_{i,t}$ is the entropy of the $t$-th token in the $i$-th response, these methods can be understood as: a **column-wise comparison**, which compares the entropy of tokens at the same position across different responses (i.e., within a column of $H$); a **row-wise comparison**, which

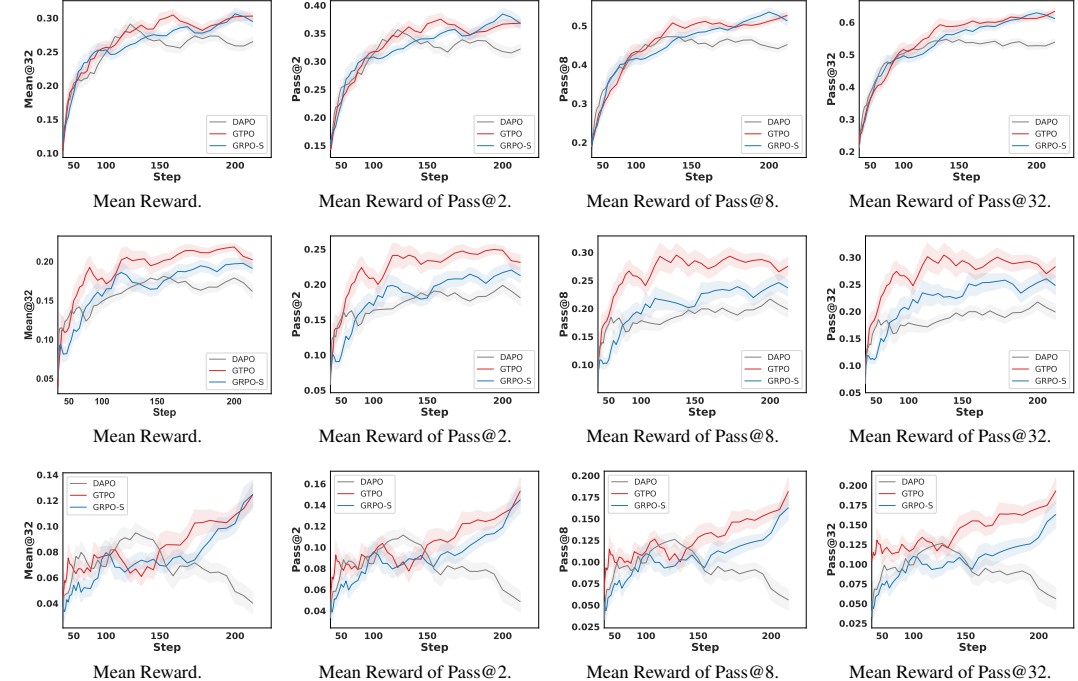

**Figure 4:** Mean reward trajectories on the test sets. **All curves are smoothed for visual clarity.** Each row corresponds to a different experimental setting: (Top) AIME 2024 with Qwen2.5-32B, (Middle) AIME 2025 with Qwen2.5-32B, and (Bottom) AIME 2025 with Qwen2.5-7B. Columns show different metrics from left to right: Mean Reward, Mean Reward of Pass@2, Pass@8, and Pass@32. For brevity and to maintain visual clarity, the corresponding results for Pass@4 and Pass@16, which exhibit similar trends, are deferred to Appendix E.3, see Fig. 7 and Fig. 8.

compares tokens at different positions within the same response (i.e., within a row of $H$); and a **matrix-wise comparison**, which compares all tokens from all responses collectively (i.e., across the entire matrix $H$). Further experimentation is needed to verify which of these approaches is superior. The conceptual difference is illustrated below:

Method 1 (Column-wise):     Method 2 (Row-wise):     Method 3 (Matrix-wise):

$$\begin{bmatrix} H_{1,1} & \dots & H_{1,t} & \dots & H_{1,l} \\ H_{2,1} & \dots & H_{2,t} & \dots & H_{2,l} \\ \vdots & \ddots & \vdots & \ddots & \vdots \\ H_{i,1} & \dots & H_{i,t} & \dots & H_{i,l} \\ \vdots & \ddots & \vdots & \ddots & \vdots \\ H_{n,1} & \dots & H_{n,t} & \dots & H_{n,l} \end{bmatrix} \begin{bmatrix} H_{1,1} & \dots & H_{1,t} & \dots & H_{1,l} \\ H_{2,1} & \dots & H_{2,t} & \dots & H_{2,l} \\ \vdots & \ddots & \vdots & \ddots & \vdots \\ H_{i,1} & \dots & H_{i,t} & \dots & H_{i,l} \\ \vdots & \ddots & \vdots & \ddots & \vdots \\ H_{n,1} & \dots & H_{n,t} & \dots & H_{n,l} \end{bmatrix} \begin{bmatrix} H_{1,1} & \dots & H_{1,t} & \dots & H_{1,l} \\ H_{2,1} & \dots & H_{2,t} & \dots & H_{2,l} \\ \vdots & \ddots & \vdots & \ddots & \vdots \\ H_{i,1} & \dots & H_{i,t} & \dots & H_{i,l} \\ \vdots & \ddots & \vdots & \ddots & \vdots \\ H_{n,1} & \dots & H_{n,t} & \dots & H_{n,l} \end{bmatrix}$$

## 5 CONCLUSION

In this paper, we addressed the fundamental challenge of coarse-grained credit assignment, a critical flaw in aligning large language models for complex reasoning. We proposed a novel framework centered on dynamic entropy weighting, which introduces two new algorithms: Group Token Policy Optimization (GTPO) for precise, token-level supervision, and a computationally efficient variant, Sequence-Level GRPO (GRPO-S). Our approach repurposes policy entropy as a proxy for model uncertainty to concentrate the learning signal at critical decision points, thereby enabling principled, fine-grained credit assignment. Extensive experiments demonstrate that our methods consistently outperform strong DAPO and GRPO baselines across multiple reasoning benchmarks, confirming the efficacy of the proposed entropy-weighting mechanism. Ultimately, our findings suggest that harnessing and directing model uncertainty is a promising frontier for developing the next generation of powerful and reliable AI systems, as demonstrated qualitatively in the case study in Appendix F.

## ETHICS STATEMENT

All contributing authors of this paper confirm that they have read and undertake to abide by the ICLR Code of Ethics. Additionally, this research has obtained approval from the Institutional Review Board (IRB) of our organization.In the conduct of this study, open-source datasets, training frameworks, and base models were utilized. A comprehensive risk assessment indicates that this research presents no potential risks. All participants involved in the project participated on a voluntary basis and did not receive any form of additional compensation.

## REPRODUCIBILITY STATEMENT

All specific details pertaining to the datasets and our experimental configurations are available in Section 3. Furthermore, the formulations and proofs corresponding to the theoretical components of this study are provided in Appendices B and C. Additionally, qualitative cases illustrating the performance of different methods are included in Appendix E. All datasets employed in this research are publicly accessible, and the associated code will be released to facilitate the reproducibility of our experimental results.

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

## A    USE OF LLMS

This research has been conducted with strict adherence to principles of authenticity and reliability. During the manuscript drafting process, large language models (LLMs) were solely utilized to refine the linguistic expression of the original research content. Specifically, their application was limited to tasks such as grammatical error verification and minor adjustments to sentence structures, with the aim of streamlining the main text without altering its core meaning.

## B    BACKGROUND AND RELATED WORK

### B.1    THE EVOLUTION OF LLM ALIGNMENT ALGORITHMS

LLM alignment techniques aim to make model behavior conform to human expectations and values. The field was initially dominated by PPO-based RLHF. This classic paradigm consists of three stages: supervised fine-tuning (SFT), reward model training, and reinforcement learning optimization. Despite its power, its process is complex, sensitive to hyperparameters, and often unstable during training. To overcome these challenges, the research community has shifted towards more direct optimization methods. Direct Preference Optimization (DPO) was a landmark work that cleverly transformed the reward maximization problem into a simple classification loss, completely bypassing explicit reward modeling and the RL process (Rafailov et al., 2023). The success of DPO has spawned a series of variants, such as ODPO (Amini et al., 2024), which considers the strength of preferences, and Preference Tuning LLMs with TRL (Hugging Face, 2024), which aims to solve overfitting, collectively advancing the RL-free alignment paradigm.

## B.2 THE RISE OF VALUE-FUNCTION-FREE POLICY OPTIMIZATION

Our work builds directly on value-function-free policy optimization methods. **Group Relative Policy Optimization (GRPO)**, introduced by DeepSeekMath, is a representative of this direction (Shao et al., 2024). The core mechanism of GRPO is: for a given prompt, sample a group of $G$ sequences from the current policy, and then use the average reward within this group as a baseline to calculate the advantage for each sequence (Kilcher, 2024). This design eliminates the need for a separate value function network, greatly reducing memory consumption and computational complexity, which has led to great success in tasks like mathematical reasoning. However, the original GRPO is also sensitive to reward noise and can be unstable during training, which has prompted subsequent research for improvements.

## B.3 A TECHNICAL COMPARISON WITH THE DAPO BASELINE

**Decoupled Clip and Dynamic sAmpling Policy Optimization (DAPO)** (Yu et al., 2025) is currently the state-of-the-art (SOTA) method for GRPO-style training in the open-source community. DAPO significantly improves the performance and stability of GRPO by introducing four key techniques: 1) **Clip-Higher**: Encourages model exploration and prevents entropy collapse by relaxing the upper bound of the PPO clipping range. 2) **Dynamic Sampling**: Filters out sample groups that are either all successful or all failures, ensuring that each training batch contains effective gradient signals, thus improving training efficiency. 3) **Token-Level Policy Gradient Loss**: A core improvement of DAPO, its objective function averages the loss over all tokens in a batch, rather than first summing within a sequence and then averaging across sequences as in the original GRPO (Yu et al., 2025). 4) **Overlong Reward Penalty**: Penalizes excessively long generated sequences to reduce reward noise.

Our work shares some motivations with DAPO. Specifically, in Appendix C, through variance analysis, it is proven that the token-level loss normalization method used by DAPO (i.e., a single average over all tokens) is statistically superior to GRPO's two-stage averaging method. This provides a theoretical basis for our adoption of a similar loss function structure. However, our core contribution is fundamentally different from DAPO's. DAPO's token-level loss addresses the normalization of the loss *calculation*, but its advantage term $\hat{A}_{i,t}$ remains **constant** for all tokens $t$ within a given sequence $i$. This means DAPO does not solve the fundamental **credit assignment problem** raised in the introduction. Our work, particularly the GTPO algorithm, directly reconstructs the reward signal itself by introducing a dynamic, non-uniform token-level reward $\tilde{r}_{i,t}$, thereby achieving true fine-grained credit assignment. In short, DAPO optimizes *how to sum the losses*, while the content of the loss terms themselves is optimized.

## B.4 ENTROPY AS A HEURISTIC FOR COGNITIVE EFFORT IN LLMS

Using model entropy as a measure of uncertainty has a long history in the machine learning field. Recent research has shown that during the reasoning process of LLMs, the entropy of the model's generated probability distribution is highly correlated with cognitive uncertainty. For example, Cheng et al. (2025) found that in successful reasoning paths, high-entropy regions often correspond to steps where the model engages in meaningful exploration and critical logical reasoning. This finding provides strong support for our use of entropy as a heuristic for credit assignment and forms the cornerstone of our methodology.

# C THEORETICAL ANALYSIS AND PROOFS

## C.1 VARIANCE COMPARISON OF TWO MEAN CALCULATION METHODS

This section provides a detailed proof to show that when estimating the mean of a random variable, directly taking the total mean of all samples (Method 2) is superior to first calculating subgroup means and then averaging them (Method 1).

Let there be a random variable $X$ with mean $\mathbb{E}[X] = \mu$ and variance $Var(X) = \sigma^2$. There are $m$ independent groups, and for the $i$-th group, $n_i$ independent and identically distributed samples are drawn to obtain the sample set $\{x_i^{(1)}, x_i^{(2)}, ..., x_i^{(n_i)}\}$.

**Method 1: First Compute Subgroup Means, Then Average**  For each subgroup $i$ ($1 \leq i \leq m$), its sample mean is computed as follows:

$$\overline{x}_i = \frac{1}{n_i} \sum_{j=1}^{n_i} x_i^{(j)} \tag{9}$$

The expectation of each $\overline{x}_i$ is $\mathbb{E}[\overline{x}_i] = \mu$, and its variance is $Var(\overline{x}_i) = \frac{\sigma^2}{n_i}$. Then, the mean of $X$ is estimated as the average of these subgroup means:

$$\hat{X}_1 = \frac{1}{m} \sum_{i=1}^{m} \overline{x}_i \tag{10}$$

The expectation of $\hat{X}_1$ is $\mathbb{E}[\hat{X}_1] = \frac{1}{m} \sum_{i=1}^{m} \mathbb{E}[\overline{x}_i] = \mu$, which is an unbiased estimator. Its variance is:

$$Var(\hat{X}_1) = \frac{\sigma^2}{m^2} \sum_{i=1}^{m} \frac{1}{n_i} \tag{11}$$

**Method 2: Directly Compute the Grand Mean of All Samples**  The total number of samples is $N = \sum_{i=1}^{m} n_i$. The mean of all samples is computed directly:

$$\hat{X}_2 = \frac{1}{N} \sum_{i=1}^{m} \sum_{j=1}^{n_i} x_i^{(j)} \tag{12}$$

The expectation of $\hat{X}_2$ is $\mathbb{E}[\hat{X}_2] = \mu$, also an unbiased estimator. Its variance is:

$$Var(\hat{X}_2) = \frac{\sigma^2}{N} = \frac{\sigma^2}{\sum_{i=1}^{m} n_i} \tag{13}$$

**Comparing $Var(\hat{X}_1)$ and $Var(\hat{X}_2)$**  According to the Arithmetic Mean-Harmonic Mean (AM-HM) inequality, for any set of positive numbers $n_1, ..., n_m$, the following holds:

$$\frac{\sum_{i=1}^{m} n_i}{m} \geq \frac{m}{\sum_{i=1}^{m} \frac{1}{n_i}} \tag{14}$$

This means the arithmetic mean is greater than or equal to the harmonic mean, with equality holding if and only if all $n_i$ are equal. Since $A \geq H$, it follows that $\frac{1}{A} \leq \frac{1}{H}$. Therefore,

$$Var(\hat{X}_2) \leq Var(\hat{X}_1) \tag{15}$$

It shows that Method 2 is statistically superior because it provides an estimator with smaller (or equal) variance. In the context of RL training for LLMs, $m$ corresponds to the number of sequences in a batch, $G$, and $n_i$ corresponds to the length of the $i$-th sequence, $|o_i|$. Since generated sequence lengths are typically different, $Var(\hat{X}_2) < Var(\hat{X}_1)$.

## C.2 UNIFYING THE OBJECTIVE FUNCTION OF GRPO AT TOKEN-LEVEL

However, the previous proof is based on the assumption that the random variables are uniformly distributed. For a more precise proof, a substitution needs to be performed on the random variables.

Continuing from the random variable $X$ above, consider the random variable $Y = f(X)$, where $f$ does not have a specific functional form, but given a value of $X$, a deterministic value of $Y$ (analogous to a neural network) can be obtained . For a specific sample $x_i^{(j)}$, the corresponding value of $Y$ can be obtained as:

$$y_i^{(j)} = f(x_i^{(j)}) = \frac{\pi_\theta(o_{i,j}|q, o_{i,<j})}{\pi_{\theta_{old}}(o_{i,j}|q, o_{i,<j})} (x_i^{(j)} - c),$$

where $c$ is a constant, corresponding to mean($\{R_i\}_{i=1}^{G}$).

If $\bar{y}_i = \frac{1}{n_i} \sum_{j=1}^{n_i} y_i^{(j)}$ is defined and use the two methods from above to estimate $[Y]$, an identical proof allows us to obtain:

$$Var(\hat{Y}_2) \leq Var(\hat{Y}_1).$$

This completes the proof, leading to the following conclusion.

**Conclusion 1.** If unifying GRPO at the token-level is considered, a single average $\frac{1}{\sum |o_i|} \sum$ should be used rather than a two-stage average $\frac{1}{G} \sum \frac{1}{|o_i|} \sum$. Therefore, the leading coefficient of GTPO, $\frac{1}{\sum |o_i|}$, is superior to that of GRPO.

Next, the $\text{mean}(\{R_i\}_{i=1}^G)$ part of the GRPO objective function is analyzed. First, the ideal state of the advantage function is known to be $A = Q - V$, where $Q$ is the action-value and $V$ is the state-value. Therefore, during sampling, Q and V need to be estimated as accurately as possible. The analysis in A.1 for $Y$ is actually an analysis of the estimation method for $Q$. DAPO has already modified this, but changing the estimation method for $V$ can be considered. The term $\text{mean}(\{R_i\}_{i=1}^G)$ is the estimate for $V$. Theoretically, there is a more accurate estimation method, which is proven below.

Currently, the way GRPO and DAPO assign rewards to each token in a sequence is by taking the reward from the last token and assigning it to all preceding tokens in that sequence. The final sampling result is equivalent to the result obtained from the following sampling method: $G$ groups of samples are sampled, where each group $o_i$ corresponds to a set of samples $\{r_i^{(1)}, r_i^{(2)}, \ldots, r_i^{(|o_i|)}\}$. The arithmetic mean for each group is then taken to get $\bar{r}_i = \frac{1}{|o_i|} \sum_{j=1}^{|o_i|} r_i^{(j)}$, and then the collected sample for each group is assumed to be $\{\bar{r}_i, \bar{r}_i, \ldots, \bar{r}_i\}$ ($|o_i|$ times).

Without confusion, the notation $\{r_i^{(1)}, \ldots, r_i^{(|o_i|)}\}$ is still used to to represent the set of samples corresponding to $o_i$, but it must be noted that the relation $r_i^{(1)} = r_i^{(2)} = \cdots = r_i^{(|o_i|)} = \bar{r}_i$ holds.

Since the problem is considered at the token-level, the $R$ (reward) in $\text{mean}(R)$ should also be at the token-level, not simply at the sequence-level. The length of the sequence (i.e., the number of tokens) cannot be ignored just because the default sample values within each sequence are identical. The reason is as follows: since the sampling of rewards here is all i.i.d., it is completely equivalent to the analysis of the random variable $X$ in A.1. Here, i.i.d. samples of $R$ are being dealt with. Let the following be set:

$$\hat{R}_1 := \frac{1}{G} \sum_{i=1}^G \bar{r}_i, \quad \hat{R}_2 := \frac{1}{\sum |o_i|} \sum_{i=1}^G \sum_{j=1}^{|o_i|} r_i^{(j)}.$$

If the reward corresponding to the last token is defined as $r_i$, the following notation consistent with GRPO is used:

$$\hat{R}_1 = \frac{1}{G} \sum_{i=1}^G r_i, \quad \hat{R}_2 = \frac{\sum |o_i| r_i}{\sum |o_i|}.$$

Based on the previous proof for $X$, it can be easily concluded that $Var(\hat{R}_2) \leq Var(\hat{R}_1)$. This completes the proof, leading to the following conclusion.

**Conclusion 2.** If unifying GRPO at the token-level is considered, $\text{mean}(\{R_i\}_{i=1}^G)$ in GRPO should be replaced with

$$\text{mean}(R) := \frac{\sum_{i=1}^G |o_i| r_i}{\sum_{i=1}^G |o_i|}.$$

### C.3 ANALYSIS OF THE SEQUENCE-LEVEL OBJECTIVE (GRPO-S)

A parallel analysis applies to the sequence-level objective function. Since the reward for incorrect sequences only affects the gradient's magnitude, not its direction, the analysis similarly centers on the reward redistribution for correct sequences. The reshaping of sequence-level rewards, $\hat{r}_i^+$, is designed to preserve the total reward across the batch. This is confirmed by the following relationship:

$$\sum_{i=1}^G \hat{r}_i^+ = (\beta_1 + \beta_2) \sum_{i=1}^G r_i \approx \sum_{i=1}^G r_i,$$

where $\beta_1 + \beta_2 \approx 1$ is set. From this property, it follows that the expected mean reward is conserved:

$$\mathbb{E}[\text{mean}(\{\hat{r}_k^+\}_{k=1}^G)] \approx \mathbb{E}[\text{mean}(\{r_k\}_{k=1}^G)].$$

This conservation property ensures that the expectation of our modified advantage function, $\mathbb{E}[\hat{A}_i^+]$, remains approximately equal to the expectation of the advantage function from the original GRPO algorithm, $\mathbb{E}[\hat{A}_i]$ ($\mathbb{E}[\hat{A}_i^+] \approx \mathbb{E}[\hat{A}_i]$). Consequently, with the entropy term detached from the gradient, the expected policy gradient of our sequence-level objective, $\mathcal{J}_{\text{GRPO-S}}(\theta)$, is approximately equivalent to that of the baseline GRPO objective:

$$\mathbb{E}[\nabla_\theta \mathcal{J}_{\text{GRPO-S}}(\theta)] \approx \mathbb{E}[\nabla_\theta \mathcal{J}_{\text{GRPO}}(\theta)].$$

Following a similar derivation to the token-level case, it is concluded that our sequence-level reward shaping preserves the convergence properties of the underlying reinforcement learning algorithm. The modification primarily serves to reshape the reward landscape to influence training dynamics, such as by reducing gradient variance, without altering the fundamental optimization direction in expectation.

### C.4 Gradient

This section briefly analyzes the expected gradient of the GTPO and GRPO-S objective function. Since the entropy term is detached during gradient calculation, it is easy to calculate that

$$\nabla_\theta J_{\text{GTPO}}(\theta) = \mathbb{E}\left[\frac{1}{\sum_{i=1}^{G}|o_i|}\sum_{i=1}^{G}\sum_{t=1}^{|o_i|} w_{i,t}(\theta)\tilde{A}_{i,t}\nabla_\theta\log\pi_\theta(o_{i,t}\mid q, o_{i,<t})\right]. \tag{16}$$

And according to policy gradient theorem, it is easy to calculate that

$$\nabla_\theta J_{\text{GRPO-S}}(\theta) = \mathbb{E}\left[\frac{1}{G}\sum_{i=1}^{G}\widehat{w}_i(\theta)\widehat{A}_i \cdot \nabla_\theta\log\pi_\theta(o_{i,t}\mid q, o_{i,<t}) \cdot \frac{1}{|o_i|}\sum_{t=1}^{|o_i|}\log\pi_\theta(o_{i,t}\mid q, o_{i,<t})\right]. \tag{17}$$

## D  Group Relative Overlong Punishment: A Heuristic for Length Control

To solve the problem of declining accuracy on simple tasks and consistent failures on complex ones, a differential penalty is proposed to be applied to the response length based on the classified difficulty of each task. Details are as follows.

First, for a given question q, a group of responses $\{o_i\}_{i=1}^{G}$ is sampled. Following the previous notation, this group is assumed to consist of n correct responses and m incorrect responses. **Easy Question** and **Hard Question** can then be distinguished.

**Easy Question:** If $\frac{n}{n+m} \geq \gamma_1$, then $q$ is called an easy question. Where $0 < \gamma_1 < 1$.

**Hard Question:** If $\frac{n}{n+m} \leq \gamma_2$, then $q$ is called a hard question. Where $\gamma_2 = 1 - \gamma_1$.

Then the **Group Relative Overlong Punishment** can be defined:

- Let $L_1^+ = \min\{|o_i|,\ 1 \leq i \leq n\}$, and let $L_2^+ = \max\{|o_i|,\ 1 \leq i \leq n\}$. Then we define

$$L^+ = \max\{\frac{L_1^+ + L_2^+}{2}, \bar{L}^+\},$$

  where $\bar{L}^+ = \frac{\sum_{i=1}^{n}|o_i|}{n}$. Then for $q$ is an **easy question**, a **Group Relative Overlong Punishment** is set for the **correct responses** as following:

$$R^+(i) = \begin{cases} -\frac{1}{2}\frac{|o_i|-L^+}{L^+} & \text{if } L^+ \leq |o_i| < 2L^+, \\ -\frac{1}{2} & \text{if } 2L^+ \leq |o_i|. \end{cases}$$

- For **hard questions**, no response length punishment is set, or the same punishment as DAPO is set. This is because the goal is to preserve the policy model's ability to produce correct answers to the greatest extent possible.

- Let $L_1^- = \min\{|o_i|,\ 1 \leq i \leq G\}$, and let $L_2^- = \max\{|o_i|,\ 1 \leq i \leq G\}$. Then define

$$L^- = \max\{\frac{L_1^- + L_2^-}{2}, \bar{L}^-\},$$

where $\bar{L}^- = \frac{\sum_{i=1}^{G} |o_i|}{G}$. For $q$ is a question that is neither easy nor hard, if $n > m$, a **Group Relative Overlong Punishment** is set for the **correct responses** as follows:

$$R^-(i) = \begin{cases} -\frac{1}{2}\frac{|o_i|-L^-}{L^-} & \text{if } L^- \leq |o_i| < 2L^-, \\ -\frac{1}{2} & \text{if } 2L^- \leq |o_i|. \end{cases}$$

If $n \leq m$, a **Group Relative Overlong Punishment** is set for the **incorrect responses** as following:

$$R^-(j) = \begin{cases} -\frac{1}{2}\frac{|o_j|-L^-}{L^-} & \text{if } L^- \leq |o_j| < 2L^-, \\ -\frac{1}{2} & \text{if } 2L^- \leq |o_j|. \end{cases}$$

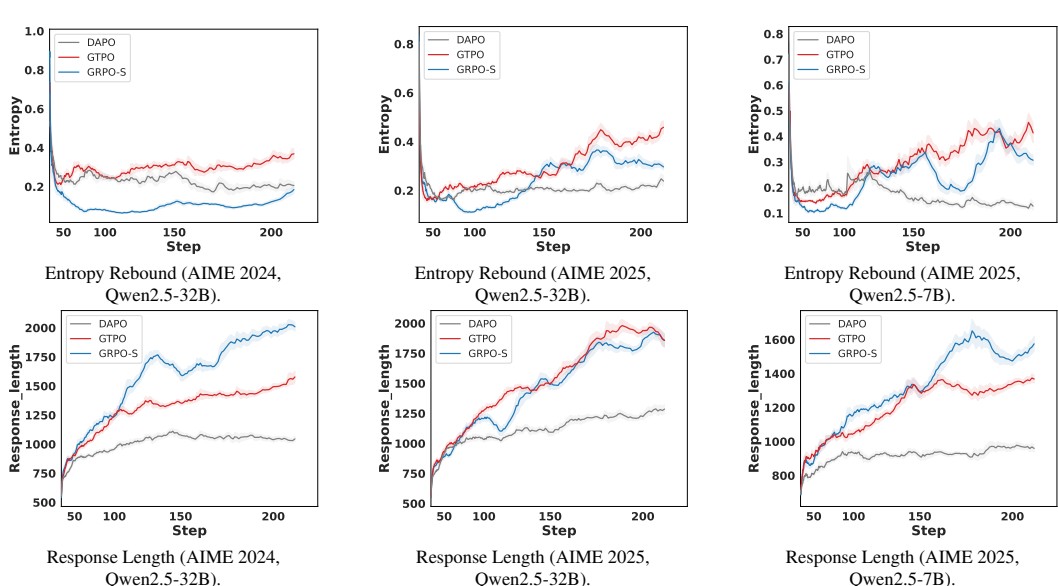

Entropy Rebound (AIME 2024, Qwen2.5-32B).

Entropy Rebound (AIME 2025, Qwen2.5-32B).

Entropy Rebound (AIME 2025, Qwen2.5-7B).

Response Length (AIME 2024, Qwen2.5-32B).

Response Length (AIME 2025, Qwen2.5-32B).

Response Length (AIME 2025, Qwen2.5-7B).

**Figure 5:** The Entropy Rebound Phenomenon and its Effect on Response Length. **Top Row:** The policy entropy trajectories for experiments on (left to right) AIME 2024 with Qwen2.5-32B, AIME 2025 with Qwen2.5-32B, and AIME 2025 with Qwen2.5-7B. Our methods (GTPO, GRPO-S) exhibit a distinct entropy rebound after an initial dip, successfully counteracting the policy collapse observed in the DAPO baseline. **Bottom Row:** The corresponding average response length trajectories. The sustained exploration enabled by the entropy rebound directly manifests as an increase in the average response length, indicating more thorough and diverse reasoning.

## E  ADDITIONAL EXPERIMENTAL RESULTS AND ANALYSIS

### E.1  ANALYSIS OF TRAINING DYNAMICS: ENTROPY REBOUND AND EXPLORATION

Figure 5 provides empirical evidence for the core mechanism of our proposed methods. The top row illustrates the "entropy rebound" phenomenon. While all methods initially exhibit a decrease in policy entropy as they learn to exploit correct strategies, the DAPO baseline's entropy continues to decline, indicating convergence to a narrow, deterministic policy—a behavior known as policy collapse. In contrast, both GTPO and GRPO-S show a distinct rebound in entropy. This is direct evidence that our entropy-weighted reward shaping successfully incentivizes the model to maintain exploration. By rewarding uncertainty on successful paths, our methods encourage the model to escape local optima and explore a more diverse set of reasoning pathways. This sustained exploration, as shown in the bottom row, directly results in longer, more detailed responses as the model attempts more thorough lines of reasoning. This establishes a clear causal link from our reward mechanism to superior problem-solving capabilities, as validated by the results in Table 1.

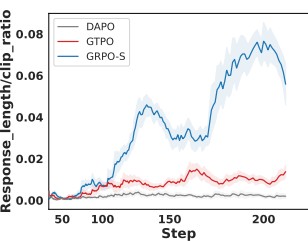 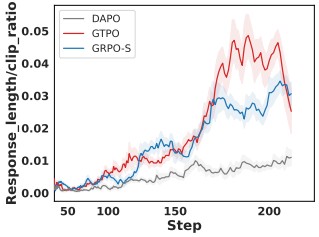 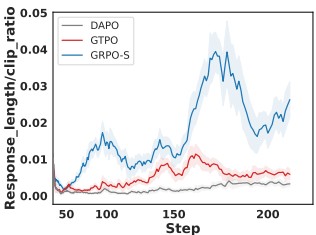

Response Length Clip Ratio (AIME 2024, Qwen2.5-32B). | Response Length Clip Ratio (AIME 2025, Qwen2.5-32B). | Response Length Clip Ratio (AIME 2025, Qwen2.5-7B).

**Figure 6:** Response Length Clip Ratio Trajectories. The plots show the fraction of generated sequences that reached the maximum length limit of 4096 tokens for experiments on (left to right) AIME 2024 with Qwen2.5-32B, AIME 2025 with Qwen2.5-32B, and AIME 2025 with Qwen2.5-7B. The consistently higher clip ratio for GTPO and GRPO-S (around 10%) compared to DAPO provides further evidence of enhanced exploration, as our methods encourage the model to generate more thorough responses that often utilize the full generation budget.

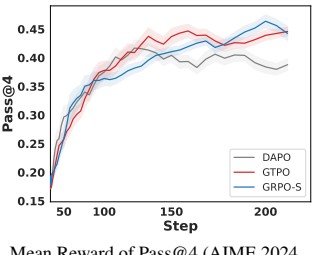 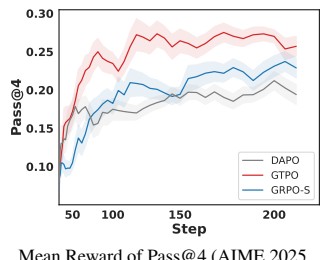 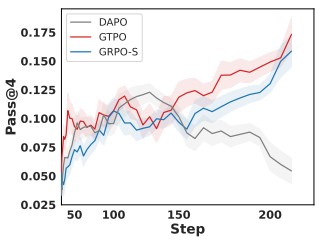

Mean Reward of Pass@4 (AIME 2024, Qwen2.5-32B). | Mean Reward of Pass@4 (AIME 2025, Qwen2.5-32B). | Mean Reward of Pass@4 (AIME 2025, Qwen2.5-7B).

**Figure 7:** Mean Reward Trajectories for Pass@4 on Test Sets. The plots show the mean reward of the top 4 generations for experiments on (left to right) AIME 2024 with Qwen2.5-32B, AIME 2025 with Qwen2.5-32B, and AIME 2025 with Qwen2.5-7B. The trends are consistent with those reported in the main paper, with GTPO and GRPO-S achieving a higher reward ceiling than the DAPO baseline.

### E.2 ANALYSIS OF GENERATION CHARACTERISTICS: RESPONSE LENGTH AND CLIPPING

Figure 6 displays the response length clip ratio, which is the fraction of generated sequences that reach the maximum token limit. The significantly higher clip ratio for GTPO and GRPO-S serves as further evidence of enhanced exploration. This indicates that our methods encourage the model to generate more elaborate and detailed reasoning chains, often exhausting the available generation budget. This contrasts with the DAPO baseline, where premature policy convergence leads to shorter, less exploratory responses.

### E.3 COMPLETE REWARD TRAJECTORIES ON TEST SETS

Figures 7 and 8 present the mean reward trajectories for Pass@4 and Pass@16, respectively. These plots complete the picture presented in Figure 3 of the main paper. The observed trends are highly consistent across all Pass@k metrics: GTPO and GRPO-S consistently achieve a higher final reward ceiling than the DAPO baseline, demonstrating the robustness of our performance gains.

### E.4 REWARD TRAJECTORIES ON TRAINING SETS

Figure 9 shows the mean reward trajectories on the training sets. These curves are crucial for evaluating sample efficiency. The plots indicate that all models, including our proposed methods and the baseline, largely converge within 210 training steps. This demonstrates that the substantial performance improvements achieved by GTPO and GRPO-S are not the result of longer training but are due to a more effective and efficient learning signal derived from our entropy-weighting mechanism.

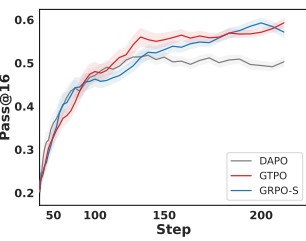 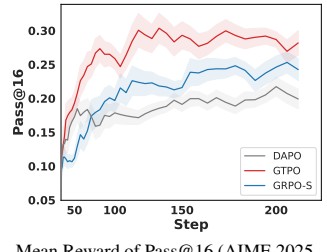 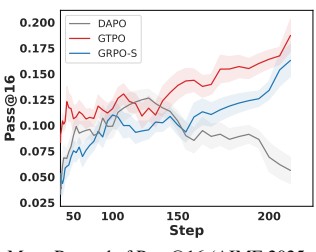

Mean Reward of Pass@16 (AIME 2024, Qwen2.5-32B). | Mean Reward of Pass@16 (AIME 2025, Qwen2.5-32B). | Mean Reward of Pass@16 (AIME 2025, Qwen2.5-7B).

**Figure 8:** Mean Reward Trajectories for Pass@16 on Test Sets. The plots show the mean reward of the top 16 generations for experiments on (left to right) AIME 2024 with Qwen2.5-32B, AIME 2025 with Qwen2.5-32B, and AIME 2025 with Qwen2.5-7B. These results further demonstrate the robust and consistent performance improvements of our methods across different evaluation metrics.

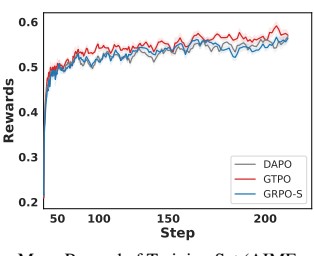 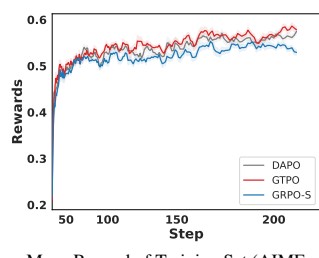 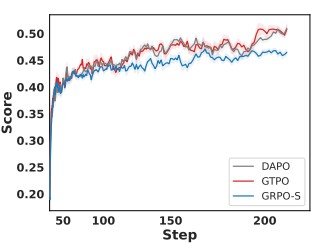

Mean Reward of Training Set (AIME 2024, Qwen2.5-32B). | Mean Reward of Training Set (AIME 2025, Qwen2.5-32B). | Mean Reward of Training Set (AIME 2025, Qwen2.5-7B).

**Figure 9:** Mean Reward Trajectories on Training Sets. The plots show the mean reward on the training data for experiments on (left to right) AIME 2024 with Qwen2.5-32B, AIME 2025 with Qwen2.5-32B, and AIME 2025 with Qwen2.5-7B. The reward curves demonstrate that all models approach convergence on the training data by approximately 210 steps, supporting the claim that the performance benefits of our methods do not come at the cost of reduced sample efficiency.

# F    QUALITATIVE CASE STUDY

**Problem.**

Alice chooses a set $A$ of positive integers. Then Bob lists all finite nonempty sets $B$ of positive integers with the property that the maximum element of $B$ belongs to $A$. Bob's list has 2024 sets. Find the sum of the elements of $A$.

**Solution of GTPO:**

For a given set $A = \{a_1, a_2, \ldots, a_n\}$, where $a_1 < a_2 < \cdots < a_n$, and for any subset $B \subset A$, the maximum element of $B$ must belong to $A$. For a fixed maximum element $a_i \in B \subset A$, the set $B$ can include any subset of the elements $\{1, 2, \ldots, a_i - 1\}$ along with $a_i$. Thus, for a given $a_i$, the number of possible sets $B$ with maximum element $a_i$ is the number of subsets of $\{1, 2, \ldots, a_i - 1\}$ plus the element $a_i$ itself. This is $2^{a_i - 1}$. Therefore, the total number of such sets $B$ is the sum of $2^{a_i - 1}$ for all $a_i \in A$, giving $\sum_{i=1}^{n} 2^{a_i - 1} = 2024$. The task is then to find a set $A = \{a_1, a_2, \ldots, a_n\}$ that satisfies the above equation. To find such $A$, we can express 2024 as a sum of powers of 2.

**Then 2024 is converted to its unique binary representation:**

$$\mathbf{2024_{10} = 11111101000_2}$$
$$\mathbf{= 2^{10} + 2^9 + \cdots + 2^6 + 2^5 + 2^3}.$$

Since the binary representation of any integer is unique, we have

$$n = 7, a_1 - 1 = 10, a_2 - 1 = 9, a_3 - 1 = 8, a_4 - 1 = 7, a_5 - 1 = 6, a_6 - 1 = 5, a_7 - 1 = 3.$$

Then $\cdots$, we get $\sum_{i=1}^{n} a_i = 4 + 6 + 7 + 8 + 9 + 10 + 11 = 55$.

Hence, we get the sum of the elements of $A$ is $\boxed{55}$.  ✓

**Solution of GRPO-S:**

**Step 1: Understanding the Sets $B$.** Denote the elements of $A$ as $\{a_1, a_2, \ldots, a_n\}$, where $a_1 < a_2 < \cdots < a_n$. According to the problem statement, we know that the maximum element of $B$ must belong to $A$.

**Step 2: Counting the Sets $B$.** For each $a_i \in A$, consider the sets $B$ where $a_i$ is the maximum element. Note that $a_i$ is the maximum element of $B$, then the other elements of $B$ can be any subset of $\{1, 2, \ldots, a_i - 1\}$. The number of subsets of $\{1, 2, \ldots, a_i - 1\}$ is $2^{a_i - 1}$. Since $a_i$ is the maximum element of $B$, $a_i \in B$, which means $B$ is nonempty. Thus, there are $2^{a_i - 1}$ possible sets $B$ for each $a_i$.

**Step 3: Total Number of Sets $B$.** Since the sets $B$ are disjoint based on their maximum elements, the total number of sets $B$ is the sum of the number of sets for each $a_i$. This gives $\sum_{i=1}^{n} 2^{a_i - 1} = 2024$.

**Step 4: Finding the Set $A$.** Then the task is to find a set $A = \{a_1, a_2, \ldots, a_n\}$ such that the sum of $2^{a_i - 1}$ for all $i$ equals 2024. This is equivalent to finding a set of exponents such that the sum of the corresponding powers of 2 equals 2024.

**Let's express 2024 in binary uniquely to identify the exponents:**

$$2024_{10} = 11111101000_2.$$

The unique binary representation tells us that:

$$n = 7, a_1 - 1 = 10, a_2 - 1 = 9, a_3 - 1 = 8, a_4 - 1 = 7, a_5 - 1 = 6, a_6 - 1 = 5, a_7 - 1 = 3.$$

Then $\cdots$, which get $\sum_{i=1}^{n} a_i = 4 + 6 + 7 + 8 + 9 + 10 + 11 = 55$.

Hence, the sum of the elements of $A$ is $\boxed{55}$. ✓

---

**Solution of DAPO:**

Denote $A$ as $\{a_1, a_2, \ldots, a_n\}$, where $a_1 < a_2 < \cdots < a_n$. For each $a_i$, the sets $B$ where the maximum element is $a_i$ can be formed by taking any subset of $\{1, 2, \ldots, a_i - 1\}$ and adding $a_i$ to it. The number of such sets is $2^{a_i - 1}$ because there are $2^{a_i - 1}$ subsets of $\{1, 2, \ldots, a_i - 1\}$. Thus, the total number of sets $B$ is the sum of $2^{a_i - 1}$ for all $a_i \in A$:

$$2024 = 2^{a_1 - 1} + 2^{a_2 - 1} + \cdots + 2^{a_n - 1}$$
$$= 2^{10} + 2^9 + \cdots + 2^3 + 2^2 + 2^0$$

This means $A$ can be chosen such that $a_i - 1$ corresponds to the exponents in the right-hand side of the above equation. Thus, it follows that:

$$a_1 = 1, \ a_2 = 3, \ a_3 = 4, \ a_4 = 5, a_5 = 6, \ a_6 = 7, \ a_7 = 8, \ a_8 = 9, a_9 = 10, \ a_{10} = 11.$$

Therefore, $\sum_{i=1}^{n} a_i = 1 + 3 + 4 + 5 + 6 + 7 + 8 + 9 + 10 + 11 = 64$.

Hence, the sum of the elements of $A$ is $\boxed{64}$. ✗ >>> 55

---

**Analysis** This case study provides a compelling qualitative illustration of the practical difference between coarse-grained and fine-grained credit assignment. The DAPO model's failure is characteristic of a system that has learned the general structure of a solution (the "template") but lacks logical rigor in its execution. Its coarse-grained reward signal (a single +1 or -1 for the entire sequence) is insufficient to penalize subtle but critical errors like the incorrect binary decomposition. The model can thus become overconfident in a flawed reasoning path. In contrast, the success of the GTPO and GRPO-S models highlights the benefit of an entropy-aware reward signal. Our methods are designed to penalize low-entropy (high-confidence) mistakes, which would directly discourage the kind of confident but incorrect decomposition made by the DAPO model. Simultaneously, by rewarding exploration in successful paths, our methods encourage a more careful and deliberate reasoning process, leading to the discovery of the correct, logically sound solution. This case demonstrates that our framework is key to moving LLMs beyond mere pattern imitation towards robust, verifiable reasoning.

