# OpenReview forum: "GTPO AND GRPO-S: TOKEN AND SEQUENCE-LEVEL REWARD SHAPING WITH POLICY ENTROPY"
_ICLR.cc/2026/Conference — Submitted to ICLR 2026_

### Official Review · Reviewer_PDqi · 2025-10-16

**Soundness:** 3
**Presentation:** 3
**Contribution:** 2
**Rating:** 4
**Confidence:** 4

**Summary:**

This paper tries to addresses the coarse-grained reward assignment challenge in reasoning by leveraging policy entropy to redistribute reward signals. The authors operate on the central hypothesis that the moments of high policy entropy within a reasoning sequence are not random noise but strong correlates of pivotal reasoning exploration. To exploit this, they propose the Dynamic Entropy Weighting framework, which reshapes rewards to be proportional to token-level or sequence-level entropy. This framework instantiates two methods: GTPO, derived from token-level entropy, and GRPO-S, based on sequence-level entropy. The methods are evaluated on the AIME 2024 and AIME 2025 mathematical reasoning benchmarks.

**Strengths:**

- This paper is generally well-written and easy to follow.

- This paper tackles the important challenge of reward sparsity in RLVR scenario. The core idea of redistributing sequence-level rewards to token-level actions based on action entropy in successful sequences is intuitively sound.

- Unlike previous dense reward modeling techniques that rely on model-based rewards and risk reward hacking, GTPO offers a more robust alternative. It derives dense rewards directly from rule-based verification, which can help mitigate this risk.

**Weaknesses:**

- **Need for Stronger Justification of the Core Hypothesis**

The proposed method is based on the hypothesis that "high-entropy moments within a reasoning sequence are signatures of pivotal junctures, not noise." The paper would be strengthened by a more rigorous justification for this claim. For example, it would be beneficial to discuss the empirical conditions under which this hypothesis holds and the situations in which it might fail.

- **Concern Regarding the Penalization of Confident Tokens**

The motivation for penalizing low entropy (or confident) tokens in incorrect trajectories requires further justification. As highlighted by prior work [1], the most confident tokens in a sequence (including unsuccessful one) are often related to syntactically correct tokens (e.g., a LaTeX equation: the token "{" after "\boxed" in ["\boxed", "{"]). It is questionable whether penalizing such tokens is meaningful, as their high confidence is tied to grammatical correctness rather than reasoning error. The authors should clarify how their method distinguishes between harmful overconfidence and valid syntactic certainty.

[1] Beyond the 80/20 Rule: High-Entropy Minority Tokens Drive Effective Reinforcement Learning for LLM Reasoning, arxiv 2025.

- **Lack of Clarity in Theoretical Guarantees (Section 2.4.2)**

The theoretical derivation in Section 2.4.2, particularly the approximations at lines 308 and 313, is confusing. The equations do not explicitly provide theoretical bounds (upper or lower) for these approximations, making it difficult to assess their validity and impact. Providing more explicit and rigorous theoretical evidence would be beneficial.

- **Limited Scope of Evaluation**

The empirical evaluation is constrained to only two mathematical reasoning benchmarks (AIME 2024/2025). This narrow scope raises concerns about the robustness and generalizability of the proposed methods. Evaluation on a wider range of tasks (code generation, commonsense reasoning, or instruction following) is necessary.

- **Missing Experimental Details for Training Data**

**Questions:**

- The captions for Figures 1 and 2 lack key details, making it difficult to distinguish between high and low-entropy tokens/sequences or to identify their corresponding reward values.

- Typo: at line 260, $\tilde{r}\_{i,t}$  $\to$ $\tilde{r}^+\_{i,t}$?

---

> ### Author Response · Authors · 2025-11-23
> **For Weaknesses 1, 2**
>
> For Weakness 1:
> We appreciate you highlighting the "Need for Stronger Justification of the Core Hypothesis." We acknowledge that further elaboration is necessary in this regard.
>
> Our current experimental results focus primarily on mathematical problems. Members of our team with mathematical backgrounds posit that the hypothesis "high entropy equates to critical nodes" aligns particularly well with mathematical reasoning. This is because a key factor in solving complex mathematical problems is making correct choices at critical decision points where options are numerous and difficult to distinguish (corresponding to high entropy).
>
> For other task types, the premise that "high entropy denotes a critical node" requires more rigorous verification. However, evidence from related literature (e.g., https://arxiv.org/abs/2506.14758) indicates a strong positive correlation between high entropy and critical nodes. To address the variation across tasks, we proposed three methods for measuring entropy levels in the paper. For tasks where steps cannot be aligned at the same time step, Method 3 is applicable. We will supplement the manuscript with this discussion and relevant references.
>
> For Weakness 2:
> Regarding the "Concern Regarding the Penalization of Confident Tokens," we believe there is a misunderstanding regarding our underlying objective. Our core goal is to mathematically induce differentiation in rewards across tokens to distinguish between the quality of steps, rather than to provide a theoretical proof of a specific theorem.
>
> The rationale for penalizing low-entropy (confident) tokens in incorrect trajectories is twofold:
>
>  1. Macroscopically: If an answer is incorrect, the motivation to improve it lies in increasing exploration and discouraging persistence in the current erroneous path.
>
>  2. Microscopically: Among incorrect responses, low entropy at a critical decision point implies the policy model confidently selected a wrong option. In such cases, higher confidence in an error indicates worse performance, warranting a relatively higher penalty. The size of the "penalty" is relative; the ultimate goal is to leverage high entropy to encourage exploration away from the current incorrect path.
>
> It is important to clarify that what we term "reward" or "penalty" is essentially a reweighting mechanism based on entropy metrics. It dynamically reallocates the original, undifferentiated reward. The feasibility of this approach relies on Methods 1, 2, and 3 proposed in the paper. These methods ensure that within a long solution process, steps of the same importance level maintain entropy values within a comparable magnitude. Specifically, the entropy of non-critical steps clusters within a similar range, as does the entropy of critical steps. This ensures that after reallocation, rewards for steps of the same level exhibit dynamic differentiation while remaining comparable, avoiding disparate comparisons between unrelated steps.
>
> Regarding the specific example you provided, it corresponds to a non-critical step.
>
>  1. If using Method 1, it implies a good alignment between step criticality and time step, so the step is compared against other non-critical steps.
>
>  2. If the alignment between criticality and time step is poor, Method 3 can be employed. This ensures critical steps are compared within their specific reward range, and non-critical steps within theirs, keeping them in distinct magnitudes.

---

> ### Author Response · Authors · 2025-11-23
> **For Weaknesses 3, 4, 5**
>
> For Weakness 3:
> Regarding the theoretical derivation in Section 2.4.2, we believe there is a misunderstanding regarding the intent of the mathematical formulation.
>
> The derivation provided is intended to demonstrate that our new algorithm does not result in significant deviations regarding the update step size. The approximation symbol ($\approx$) represents a simplified mathematical expression indicating that the average direction of the gradient update remains consistent. Specifically, the approximation you questioned implies a difference of a multiplicative factor close to 1 (e.g., a scalar scaling), which does not affect the overall structural integrity. The core point is to illustrate that the new algorithm continues to approximate the same theoretical extremum of reinforcement learning as GRPO.
>
> In practice, implementation involves technical measures such as clipping. Given the numerous approximations inherent in model settings during gradient updates, guaranteeing that these approximations yield effects identical to theoretical values is challenging.
> Currently, there is no definitive method to mathematically prove that such approximations are entirely "reasonable," yet they are necessary because strict theoretical processes are often intractable in real-world implementations.
>
> Regarding the equations in lines 308 and 313, these are straightforward calculations based on parameter settings. For instance, with specific parameters, the calculation might yield $0.9+0.1=1$ or $0.9+0.2=1.1\approx 1.$ Consequently, providing a theoretical upper bound is neither necessary nor feasible, as the range of these parameters is determined empirically through experimentation.
>
> For Weakness 4:
> We appreciate you pointing out the "Limited Scope of Evaluation." We are currently preparing these experiments and plan to include the new results in the future version of our work.
>
> For Weakness 5:
> Thank you for highlighting the "Missing Experimental Details for Training Data."
>
> In fact, all our experimental details—including the training data—are identical to those in DAPO. Our sole modification was to the relevant code based on the new algorithmic expression. To clarify the source and nature of the training data, we quote the description directly from the DAPO paper:
>
> "Our dataset is sourced from the web and official competition homepages through a combination of web scraping and manual annotation. The answers of math dataset typically come in a variety of formats, such as expression, formula and number, which makes it challenging to design comprehensive rules to parse them. To provide accurate reward signals using rules and minimize errors introduced by formula parsers, inspired by AIME, we select and transform the answers into integers, which are easy to parse."

---

> ### Author Response · Authors · 2025-11-23
> **For Questions 1, 2**
>
> For Question 1:
> Thank you for pointing out the lack of clarity in the figures. We agree that they need to be better explained and improved to serve their purpose—enabling readers to gain a high-level understanding of our work through the illustration alone. We will modify the figures accordingly in the upcoming revision.
>
> For Question 2:
> We appreciate you pointing out this typo. It should be $ \tilde{r}^+_{i,t} $.

---

### Official Review · Reviewer_m2g1 · 2025-10-29

**Soundness:** 2
**Presentation:** 2
**Contribution:** 2
**Rating:** 2
**Confidence:** 4

**Summary:**

This paper studies the sparse reward challenge in reinforcement learning when training large language models. To tackle this, the authors proposed Group Token Policy Optimization (GTPO), which assigns an entropy-weighted reward to each token, and the analogous algorithm Sequence-Level GRPO (GRPO-S).

**Strengths:**

1. The method is well motivated and easy to understand or implement.
2. The experiments show a clear advantage over GRPO and DAPO.
3. The authors discussed future works and limitations of the paper in detail.

**Weaknesses:**

1. The mathematical derivation of why we have Equations 3 and 5 is unclear. Why does this give us a better policy gradient in theory?
2. The authors tried to give some theoretical guarantees in the paper. However, all the results are in the appendix. I'm expecting at least the theorems themselves to appear in the main text. Besides, it is unclear to me why the proposed PG is unbiased. The 'approximately equal to' statement is not a formal statement that can act as a theoretical guarantee.
3. It is also unclear to me why adding entropy reduces the PG variance. I'm even more confused after reading Appendix C.2. It seems to show that directly computing the mean of all samples enjoys a smaller variance than 'First Compute Subgroup Means, Then Average'. What is this conclusion's relationship with the Dynamic Entropy Weighting method?
4. Can the improvements over GRPO/DAPO in the experiments be attributed to the different averaging method described above, instead of coming from the entropy design?
5. The novelty is also limited. Entropy is known to be an important metric for RLVR. The specific entropy design might be new, but due to the lack of a clear math motivation, why we need this design instead of other entropy-related designs is unclear.

Minor: Figure 1 is uninformative. What are the orange bars? We also cannot tell the differences between (c) and (d) from the figures themselves.

**Questions:**

See weakness.

---

> ### Author Response · Authors · 2025-11-23
> **For Weaknesses 1, 2, 3**
>
> For Weakness 1:
> Regarding the mathematical derivation of Eq. 3 and Eq. 5, we believe there is a misunderstanding regarding the nature of these equations. We represent Eq. 3 as the design of our novel reward mechanism, while Eq. 5 denotes the resulting objective function; they are not intended as a derivation of a pre-existing theoretical result.
>
> The motivation behind this design is to address the problem of coarse-grained credit assignment. Based on the hypothesis presented in our paper—that "entropy levels are positively correlated with the criticality of reasoning steps"—we utilize entropy to derive new weights for reallocating the original reward. We do not claim to derive a theoretically superior Policy Gradient (PG) in a vacuum; rather, experimental results demonstrate that our algorithm achieves substantial improvements over DAPO and GRPO, particularly on difficult mathematical problems requiring long reasoning processes.
>
> For Weakness 2:
> Regarding the unbiased nature of the PG, we wish to clarify the intent of our mathematical derivation. The derivation is provided to demonstrate that our new algorithm does not induce significant deviations in the update step size. The approximation symbol ($\approx$) is a simplified mathematical expression used to illustrate that the average direction of the gradient update remains consistent. Our core argument is that we are still approximating the same theoretical optimum of reinforcement learning as GRPO; the distinction lies in which algorithm can more effectively approximate this optimum.
>
> In practice, implementation involves technical measures such as clipping. Given the numerous approximations inherent in model settings during gradient updates, guaranteeing that these approximations yield effects identical to theoretical values is challenging. Currently, strictly proving the "reasonableness" of such approximations is mathematically intractable, which is why approximations are used in the first place.
>
> The mathematical analysis in the Appendix serves a specific purpose: to motivate the irrationality of the GRPO algorithm by showing—via a brief proof—that including response length reduces the variance of the objective function. This is counter-intuitive because a correct response may contain many invalid tokens, yet GRPO assigns them all a reward of 1. This section is motivational rather than central to the primary contribution, and thus we relegated it to the Appendix to preserve space for the two proposed algorithms.
>
> For Weakness 3:
> Regarding the comment "adding entropy reduces the PG variance," we believe there is a misunderstanding. We have never claimed that "entropy reduces variance."
>
> The content in Appendix C.2 details the process of modifying the mean and standard deviation calculations and motivates our modification of the GRPO objective function. We argue that the original GRPO objective has structural defects due to coarse-grained credit assignment. The appendix demonstrates that incorporating response length reduces the gradient variance in GRPO, which we argue is unreasonable (as longer responses are not necessarily better). In summary, the Appendix outlines the thought process leading to our new mean/std calculation method—which can indeed reduce variance—but we do not attribute this variance reduction to the introduction of entropy itself.

---

> ### Author Response · Authors · 2025-11-23
> **For Weaknesses 4, 5**
>
> For Weakness 4:
> Regarding the distinction between "entropy design" and "the different averaging method," we believe a clarification is necessary. The proof in Appendix C.2 is intended to demonstrate the irrationality of the GRPO algorithm by showing that its variance is high under the original reward design, whereas considering response length (incorrectly) lowers it.
>
> To address your concern directly: we conducted an ablation study on the averaging method described in C.2 in isolation. The results showed no significant improvement or degradation in performance, although stability was noticeably improved (we can add these results to the revision if needed). Therefore, the performance gains observed in our experiments are attributed to the novel "entropy design." This design creates dynamic differentiation in rewards across tokens, directly addressing the defects of the GRPO algorithm discussed throughout the paper.
>
> For Weakness 5:
> Regarding the "novelty" of our work, we respectfully disagree with the assessment. While the concept of entropy is well-established, our method of utilizing entropy is novel. We employ entropy as a metric to derive a new weighting mechanism that dynamically reallocates the original, non-differentiated rewards.
>
> The feasibility of this entropy-based reweighting relies on Methods 1, 2, or 3 proposed in the paper. These methods ensure that within a long solution process, steps of the same importance level maintain entropy values within a comparable magnitude. This means the entropy of non-essential steps falls within a similar range, as does the entropy of critical steps. Consequently, after reallocation, rewards for steps of the same level exhibit dynamic differentiation while remaining comparable, avoiding disparate comparisons between unrelated steps.
>
> Furthermore, our improvement is a fundamental algorithmic refinement from a mathematical perspective. It encourages the policy model to autonomously explore superior paths and self-regulate the update step size to achieve a higher performance ceiling. This is distinct from the crude addition of an entropy term to simply increase or decrease step size. We focus on mathematical problems because our ultimate goal is for the policy model to autonomously judge the quality of reasoning steps. While we cannot claim to have fully achieved this goal yet, our results on AIME 2024 and 2025 indicate that this new direction—generating grounded, dynamic reward differentiation—is viable and promising.

---

> ### Author Response · Authors · 2025-11-23
> **Again for Weakness 5**
>
> Regarding the "novelty" of our work, we believe that the discussion should not be limited to the concept of entropy alone; rather, the more critical issue is the problem of token-level or segment-level credit assignment. We first examined the following prior works on token-level or segment-level credit assignment:
>
> We have closely examined related works such as R-PRM (which constructs reward signals by analyzing the rationality of reasoning chains step-by-step, replacing coarse 'final-answer-only' rewards); VisualPRM (which provides the problem and the first step, asking the model to judge correctness [value-based] and relative quality compared to the previous step [comparison-based], thereby teaching the model to associate specific steps with specific scores); StepGRPO (which combines StepRAR for step accuracy and StepRVR for step formatting); Guided GRPO (which provides partial reasoning and tasks the model with completing the sequence); PRIME (which utilizes implicit process rewards by calculating token-level rewards via the ratio of tokens in the reward model versus the reference model outputs); VersaPRM (which batch-generates Chain-of-Thought [CoT] paths, scores them with a larger model to create large-scale step-wise labels for fine-tuning a PRM, and uses aggregation strategies to generate a final process reward); GM-PRM (which refines feedback signals from simple 'incorrect answer' to specific step-wise diagnosis by having the Reward Model generate specific error causes and corrected steps); SPRO (which employs implicit process rewards by calculating the ratio of the policy to the reference as self-guidance—normalized as a token-level reward—without training a separate reward model); and EduFlow (which uses 'step+score+label+explanation' formatting for curriculum learning, applies Monte Carlo Tree Search [MCTS] for immediate pruning during rollout, and combines process and outcome scores during the RL phase).
>
> However, the improvements in these works primarily lie at a technical or engineering level—such as model parameter tuning—or rely on subjective, task-specific standards to define new token-level rewards. In contrast, we aim to improve the underlying algorithm purely from a mathematical perspective, enabling the policy model to autonomously explore superior paths and thereby achieve a higher performance ceiling.
>
> Returning to the perspective of entropy, although the concept of entropy is well-established and widely used, our application of it is novel: we utilize an entropy-based metric to derive a new weighting mechanism that dynamically allocates the originally undifferentiated rewards. Aside from the parameters introduced by this new algorithm, we have made no technical modifications to the model structure itself. Our focus on mathematical problems stems from our ultimate goal: to fully enable the policy model to independently evaluate the quality of solution steps. While we have not yet fully achieved this goal, we have successfully induced dynamic differentiation in rewards across different steps, even if a perfect distinction between 'good' and 'bad' steps remains to be realized.

---

> ### Author Response · Authors · 2025-11-23
> **For Figure 1**
>
> Thank you for pointing out the lack of clarity in the figure. We agree that it needs to be better explained and improved to serve its purpose—enabling readers to gain a high-level understanding of our work through the illustration alone. We will modify the figure accordingly in the upcoming revision.

---

### Official Review · Reviewer_jLju · 2025-10-29

**Soundness:** 2
**Presentation:** 2
**Contribution:** 2
**Rating:** 2
**Confidence:** 4

**Summary:**

This paper addresses a central limitation in reinforcement learning (RL) for large language model (LLM) reasoning—coarse-grained credit assignment, where a uniform reward is assigned to all tokens in a generated sequence. The authors propose Dynamic Entropy Weighting, a novel mechanism that uses policy entropy as a signal of “cognitive effort,” repurposing it for fine-grained reward shaping. Two new algorithms are introduced: Group Token Policy Optimization (GTPO), which performs entropy-weighted reward assignment at the token level, and Sequence-Level Group Relative Policy Optimization (GRPO-S), which applies entropy-based modulation at the sequence level. Both methods are designed as value-function-free extensions to the GRPO framework, maintaining convergence guarantees while reducing gradient variance. Experimental results on AIME 2024 and 2025 benchmarks using Qwen2.5 models demonstrate significant performance gains over DAPO and GRPO baselines, especially in small model settings.

**Strengths:**

- The proposed solution is simple yet effective.
- The paper is easy to follow.

**Weaknesses:**

- The major concern is the experimental evaluation. The experiments are only conducted on two Qwen2.5 series models and AIME benchmarks only. The authors are suggested to conduct more experiments on other base models (e.g., Llama and DeepSeek-R1-Distill series, etc.) and other benchmarks (e.g., AMC23, Minerva Math, OlympiadBench, LiveCodeBench) to validate the effectiveness and generalization of the proposed methods.
- The proposed methods introduce four hyperparameters (i.e., $\alpha_1$, $\alpha_2$, $\beta_1$ and $\beta_2$), which make it difficult to tune the methods in practice.
- The authors are suggested to provide more analysis on the sensitivity of these hyperparameters. The authors only conduct ablation study using three sets of these hyperparameters. However, there are still many other combinations of these hyperparameters. It is unclear how sensitive the proposed methods are to these hyperparameters, e.g., what if $\alpha_2$ and $\beta_2$ are set to 0.05.

**Questions:**

Please refer to the weaknesses above.

---

> ### Author Response · Authors · 2025-11-23
> **For all Weaknesses**
>
> For Weakness 1, 3:
> We thank the reviewer for these valuable suggestions. We are currently in the process of conducting these experiments and plan to incorporate the results in the future version of our work.
>
> For Weakness 2:
> Regarding this issue, we believe there is a misunderstanding regarding the parameter complexity. The reason for the presence of four parameters is that we present two distinct algorithms; however, each algorithm requires only two parameters, and their roles are clearly defined and intuitive.
> Taking $\alpha_1$ and $\alpha_2$ as examples:
>
> 1. $ \alpha_1$ (Base Reward): This parameter is set close to 1 (e.g., 0.8, 0.9, 1, 1.1). Its primary function is to serve as the main component of the reward, distinguishing whether the current token belongs to a correct response or an incorrect one.
>
> 2. $\alpha_2$ (Differentiation Term): This parameter is set close to 0 (e.g., 0.1, 0.2). Its purpose is to introduce reward differentiation among different tokens.
>
> Based on our experiments, the suitability of the selected parameters can be readily confirmed from the model's performance in the early stages of training. Therefore, the tuning process is straightforward and not difficult.

---

### Official Review · Reviewer_uVoV · 2025-11-02

**Soundness:** 2
**Presentation:** 2
**Contribution:** 2
**Rating:** 2
**Confidence:** 3

**Summary:**

The paper addresses the coarse-grained credit assignment problem in reinforcement learning for large language models, where existing methods (e.g. GRPO) give the same reward to every token based only on final outcome. To enable fine-grained credit assignment in long-chain reasoning tasks, the authors introduce Dynamic Entropy Weighting, which uses a model’s policy entropy as a proxy for cognitive effort to shape rewards at a finer level. Two algorithms are proposed: Group Token Policy Optimization (GTPO), assigning an entropy-weighted reward to each token, and Sequence-Level GRPO (GRPO-S), scaling an entire sequence’s reward by its average entropy. The paper formalizes coarse credit assignment as a bottleneck and hypothesizes that high-entropy decisions indicate pivotal reasoning steps to be reinforced. A theoretical analysis shows the new reward shaping preserves the policy’s optimality while reducing variance. Experiments on mathematical reasoning benchmarks (AIME 2024/2025) demonstrate that GTPO and GRPO-S significantly outperform baselines (DAPO and GRPO), achieving higher Pass@k and mean scores across model scales. Key contributions include introducing per-token reward shaping via entropy, a sequence-level variant for efficiency, and empirical validation of improved learning signals for complex reasoning.

**Strengths:**

1. The paper clearly identifies coarse-grained credit assignment as a fundamental limitation in current RL fine-tuning of LLMs.
2. Instead of treating policy entropy as mere “uncertainty,” the paper repurposes it as a proxy for cognitive effort, rewarding high-entropy decisions in correct answers and penalizing overconfident errors.

**Weaknesses:**

1. The paper discussion of related literature is limited in scope, potentially missing important context. Many earlier works have attempted token-level rewards or alternative credit assignment heuristics, the paper should contrast with them. Besides, given the focus on entropy, one might expect references to prior uses of entropy or uncertainty in exploration or credit assignment.
2. Some aspects of the technical presentation suffer from notation inconsistencies or ambiguity, which could confuse readers.
* For example, the formula for the shaped negative sequence reward uses $|o_j|$ in the product $\prod_{k=1}^{|o_j|}(1/\hat{H}_k)^{1/|o_j|}$, whereas in the original definition (Eq. (6) in Sec. 2.3) the normalization for negative sequences was over the count of unsuccessful sequences $m$. This suggests a likely typo or misuse of $|o_j|$ (sequence length) where $m$ (number of sequences) was intended, but the text does not clarify it. Such a notation error can be misleading, as it conflates two different concepts (sequence length vs. batch size of negative examples).
* Similarly, in the token-level formula,  $\tilde{r}{j,t}^{-}$ uses a normalization over $\sum{k=1}^{m}(1/H_{k,t})$ in Eq. (3), but the geometric mean version in Sec. 2.4 writes $\prod_{k=1}^{|o_j|}1/H_{k,t}$ , again inconsistent in indexing. These minor inconsistencies indicate that the math notation was not thoroughly proofread for consistency with the earlier definitions. Readers might have to infer the intended meaning (e.g., assume the product is actually over all $m$ negative sequences at timestep $t$) which disrupts clarity.
3. The conservation of total reward assumption (Eq. 9) and, by extension, all subsequent theoretical claims that depend on it. This includes the claim that the expected mean reward is unchanged (Eq. 10), the claim that the expected advantage is preserved (Section 2.4.2, Para 4), and the final conclusion that the expected policy gradient direction is preserved (Section 2.4.2, Para 5). The theory presented does not apply to the method that was experimentally evaluated.

**Questions:**

1. Which prior works on token-level or segment-level credit assignment have you considered, and how does your approach differ from or improve upon them?
2. How consistently does high policy entropy truly correspond to “pivotal” reasoning steps that deserve more credit?
3. Do you expect GTPO/GRPO-S to help in other domains beyond math reasoning?
4. GTPO in particular introduces per-token calculations. How much slower or more memory-intensive was training with GTPO compared to GRPO? Is this method feasible for very long sequences (since entropy must be computed at each position)? Also, did you encounter any training instabilities when combining PPO with these dynamically changing rewards?

---

> ### Author Response · Authors · 2025-11-23
> **For Weaknesses 2, 3**
>
> For Weakness 2:
> We acknowledge that we overlooked some notational errors in the manuscript, and we sincerely appreciate you pointing them out. We will correct these in the revised version.
>
> For Weakness 3:
> Regarding the total reward assumption (Eq. 9), we believe there is a misunderstanding regarding our intent. The derivation provided is intended to demonstrate that our new algorithm does not result in significant deviations regarding the update step size. The approximation symbol ($\approx$) represents a simplified mathematical expression indicating that the average direction of the gradient update remains consistent. Specifically, the approximation you questioned implies a difference of a multiplicative factor close to 1 (e.g., a scalar scaling), which does not affect the overall structural integrity. The core point is to illustrate that the new algorithm continues to approximate the same theoretical extremum of reinforcement learning as GRPO.
>
> In practice, the implementation involves technical measures such as clipping. Given the numerous approximations inherent in model settings during gradient updates, guaranteeing that these approximations yield effects identical to theoretical values is challenging. Currently, there is no definitive mathematical method to strictly prove an approximation is entirely "reasonable," as approximations are often necessitated precisely because the strict theoretical process is intractable in real-world implementations.

---

> ### Author Response · Authors · 2025-11-23
> **For Questions 2, 3, 4**
>
> For Question 2:
> Regarding this question, we first note that data experiments and results from (https://arxiv.org/abs/2506.14758) demonstrate a positive correlation between high entropy and the criticality of reasoning steps in correct responses. Furthermore, as our focus is on mathematical problems, we aim for the policy model to autonomously learn to distinguish between the quality of different steps. We believe a key factor in this learning process is exposing the model to a variety of solution paths.
>
> Consequently, we posit that amplifying high-entropy signals is a viable strategy, which is supported by our experimental results. This is because, in complex, long-chain mathematical reasoning, the steps between critical decision points are relatively straightforward. Analogously, a student proficient in mathematics solves difficult problems primarily by making correct selections at critical decision points where choices are numerous and difficult to distinguish.
>
> For Question 3:
> Current experimental results focus primarily on mathematical problems. For other task types, the premise that "high entropy denotes a critical node" requires more rigorous verification. However, data from other literature suggests a strong positive correlation between high entropy and critical nodes. Accordingly, we have provided three methods for measuring entropy levels in the paper. For tasks that cannot be aligned at the same time step t, Method 3 can be utilized. We will expand on this discussion in the subsequent revision to address this broader applicability.
>
> For Question 4:
> Experimental evidence confirms the feasibility of GTPO and GRPO-S. First, the dataset selected consists of difficult mathematical problems requiring lengthy solution processes. As shown in the response length figure in the Appendix, response lengths under GTPO and GRPO-S are relatively longer than the baseline, with a truncation rate approaching 10%. Even under these conditions, our metrics show significant improvement over the current SOTA, DAPO.
>
> Second, regarding efficiency, the number of steps to reach peak performance did not increase and was even lower than the baseline in some cases. Additionally, while we did not explicitly benchmark wall-clock time, concurrent runs with the baseline showed no significant temporal difference. In summary, the computational overhead of entropy calculation does not undermine the algorithm's feasibility. It is worth noting that when reallocating rewards for incorrect answers, instability can occur, requiring technical interventions such as clipping to ensure stability.

---

> ### Author Response · Authors · 2025-11-23
> **For Question 1**
>
> For Question 1:
> Thank you for this reminder. Previously, we believed our incorporation of entropy was distinct from traditional methods, and our primary goal was to present a novel improvement direction for the GRPO algorithm; consequently, we did not initially include comparisons with other entropy-based works. We acknowledge this oversight and will include relevant comparisons in the revision.
>
> In fact, we have closely examined related works such as R-PRM (which constructs reward signals by analyzing the rationality of reasoning chains step-by-step, replacing coarse 'final-answer-only' rewards); VisualPRM (which provides the problem and the first step, asking the model to judge correctness [value-based] and relative quality compared to the previous step [comparison-based], thereby teaching the model to associate specific steps with specific scores); StepGRPO (which combines StepRAR for step accuracy and StepRVR for step formatting); Guided GRPO (which provides partial reasoning and tasks the model with completing the sequence); PRIME (which utilizes implicit process rewards by calculating token-level rewards via the ratio of tokens in the reward model versus the reference model outputs); VersaPRM (which batch-generates Chain-of-Thought [CoT] paths, scores them with a larger model to create large-scale step-wise labels for fine-tuning a PRM, and uses aggregation strategies to generate a final process reward); GM-PRM (which refines feedback signals from simple 'incorrect answer' to specific step-wise diagnosis by having the Reward Model generate specific error causes and corrected steps); SPRO (which employs implicit process rewards by calculating the ratio of the policy to the reference as self-guidance—normalized as a token-level reward—without training a separate reward model); and EduFlow (which uses 'step+score+label+explanation' formatting for curriculum learning, applies Monte Carlo Tree Search [MCTS] for immediate pruning during rollout, and combines process and outcome scores during the RL phase).
>
> However, the improvements in these works primarily lie at a technical or engineering level—such as model parameter tuning—or rely on subjective, task-specific standards to define new token-level rewards. In contrast, we aim to improve the underlying algorithm purely from a mathematical perspective, enabling the policy model to autonomously explore superior paths and thereby achieve a higher performance ceiling.
>
> Although the concept of entropy is well-established and widely used, our application of it is novel: we utilize an entropy-based metric to derive a new weighting mechanism that dynamically allocates the originally undifferentiated rewards. Aside from the parameters introduced by this new algorithm, we have made no technical modifications to the model structure itself. Our focus on mathematical problems stems from our ultimate goal: to fully enable the policy model to independently evaluate the quality of solution steps. While we have not yet fully achieved this goal, we have successfully induced dynamic differentiation in rewards across different steps, even if a perfect distinction between 'good' and 'bad' steps remains to be realized.
>
> Our original intention was to introduce fundamental concepts from computer science and mathematics to refine the underlying algorithmic expression, thereby addressing the issue of coarse-grained credit assignment. This allows for dynamic reward differentiation across tokens, ultimately using entropy as a medium to optimize the GRPO reward. While we utilize the concept of entropy, our approach—based on amplifying high-entropy signals to propose a new reward allocation mechanism—is fundamentally different from prior methods. Therefore, our literature review was initially constrained to GRPO-related works and literature supporting the high-entropy signal amplification hypothesis (e.g., https://arxiv.org/abs/2506.14758). We will supplement the manuscript with additional references regarding entropy.

---

### Meta-Review · Area_Chair_tvL7 · 2026-01-07

**Summary:**

This paper proposes Dynamic Entropy Weighting, a framework that addresses coarse-grained credit assignment in RL for LLM reasoning. The rough idea of this paper is to use the entropy to reshape (or token-level tweak) the trajectory level reward signals, and proposed two new algorithms, GTPO and GRPO-S. The reviewers raise several major concerns on experiments, theoretical supports, and novelty, and most of them are not addressed during rebuttal. This leads to the decision of rejection.

**Reviewer Concerns:**

During rebuttal period, addressed concerns are mostly on the writing side, including notations, related works, and clarification on existing experiments. Most major concerns are still outstanding, including limited experiments, theoretical supports, and novelty.

**Reviewer Scores:**

Reviewer uVoV: 70% = 2, 30% -> 4.
Reviewer jLju: 70% = 2, 30% -> 4.
Reviewer m2g1: 50% = 2, 50% -> 4.
Reviewer PDqi: 100% = 4.

---

### Decision · Program_Chairs · 2026-01-26

Reject